# Language network lateralization is reflected throughout the macroscale functional organization of cortex

Loïc Labache [1] ✉, Tian Ge[2,3,4], B. T. Thomas Yeo [5,6,7,8,9] & Avram J. Holmes [1,10,11,12] ✉

Hemispheric specialization is a fundamental feature of human brain organization. However, it is not yet clear to what extent the lateralization of specific cognitive processes may be evident throughout the broad functional architecture of cortex. While the majority of people exhibit left-hemispheric language dominance, a substantial minority of the population shows reverse lateralization. Using twin and family data from the Human Connectome Project, we provide evidence that atypical language dominance is associated with global shifts in cortical organization. Individuals with atypical language organization exhibit corresponding hemispheric differences in the macroscale functional gradients that situate discrete large-scale networks along a continuous spectrum, extending from unimodal through association territories. Analyses reveal that both language lateralization and gradient asymmetries are, in part, driven by genetic factors. These findings pave the way for a deeper understanding of the origins and relationships linking population-level variability in hemispheric specialization and global properties of cortical organization.

A primary architectural feature of the human brain is its homotopy, with each hemisphere exhibiting broadly comparable spatial organization in terms of cytoarchitecture, macroscopic anatomy, and associated large-scale functional systems[1–5]. Despite this fundamentally symmetrical plan, common to the vast clade of animals known as the bilateria[6], the presence of functional asymmetries have been a leading principle of human evolution[7] and, more broadly, the organization of the metazoan nervous system[6]. The hemispheric specialization of a range of specific functions has been well characterized. One of the most widely investigated is the left-lateralized high-order language network encompassing aspects of the anterior and posterior cortices[8]. However, while the lateralization of brain functions and associated behaviors has fascinated neuroscientists for over a century[9,10], the origins, mechanisms, and consequences of hemispheric specialization are still largely unknown[11–15]. In this regard, the extent to which the asymmetrical organization of discrete processes may be evident throughout the macroscale functional organization of the cortical sheet remains an open question[16].

[1]Department of Psychology, Yale University, New Haven, CT 06520, US. [2]Psychiatric and Neurodevelopmental Genetics Unit, Center for Genomic Medicine, Massachusetts General Hospital, Boston, MA 02114, US. [3]Center for Precision Psychiatry, Department of Psychiatry, Massachusetts General Hospital, Boston, MA 02114, US. [4]Stanley Center for Psychiatric Research, Broad Institute, Cambridge, MA 02142, US. [5]Department of Electrical and Computer Engineering, Centre for Sleep and Cognition, National University of Singapore, Singapore, SG 119077, Singapore. [6]Department of Electrical and Computer Engineering, Centre for Translational Magnetic Resonance Research, National University of Singapore, Singapore, SG 119077, Singapore. [7]N.1 Institute for Health, National University of Singapore, Singapore, SG 119077, Singapore. [8]Martinos Center for Biomedical Imaging, Massachusetts General Hospital, Charlestown, MA 02129, US. [9]National University of Singapore Graduate School for Integrative Sciences and Engineering, National University of Singapore, Singapore, SG 119077, Singapore. [10]Department of Psychiatry, Yale University, New Haven, CT 06520, US. [11]Wu Tsai Institute, Yale University, New Haven, CT 06520, US. [12]Department of Psychiatry, Brain Health Institute, Rutgers University, Piscataway, NJ 08854, US. ✉e-mail: loic.labache@yale.edu; avram.holmes@rutgers.edu

The detailed anatomical study of the brain systems supporting language began through the post-mortem examination of patients with acquired brain injuries and aphasias. These seminal studies revealed a set of interconnected regions in the anterior and posterior cortices of the left hemisphere that underpin healthy language functioning[17], including Broca's area within the inferior frontal gyrus adjacent to the somato/motor network and Wernicke's area within the posterior superior temporal cortex. The presence of this left lateralized system has been supported by converging evidence from in vivo imaging studies of language function in healthy populations[18] and, more recently, data-driven algorithms that parcelate cortex into discrete functional networks across a variety of task contexts[8,19]. Critically, however, the left-hemispheric dominance of the language system is not fixed across development or ubiquitous in the general populations, where atypical organization has been observed[20–22]. Although some anatomical and functional hemispheric asymmetries appear early in human development[23], language is distributed symmetrically in children, with lesions to either hemisphere resulting in an equal likelihood of associated deficits[24]. From early to late adolescence, there is a gradual transition to left-hemisphere dominance in the majority of the population[14], with atypical language organization evident in ~10 percent of individuals[25,26]. This flipped profile of a right hemisphere language system is more likely to be observed in left-handed individuals[27], although not specific to this group. In right-handed adults, 2% to 8% show a dominance reversal[28]. However, the exact mechanisms of brain lateralization are still largely unknown, as are the associated consequences on broader properties of brain organization.

The cerebral cortex is comprised of a dense tapestry of areal units embedded in corresponding processing streams and housed within associated large-scale functional networks[29,30]. The topographic organization of this complex interdigitated architecture is evident in the presence of functional gradients that situate discrete networks along continuous spectra[31]. The spatial arrangement of areal parcels along these global gradients, for instance, along a principal gradient anchored on one end by the unimodal (somatosensory/motor and visual) regions and the other by the cortical association areas that underpin complex cognition[32], reflect a fundamental property of brain organization[33,34]. Converging evidence for these macroscale gradients has been established through in vivo imaging measures of function, anatomy[35], and areal allometric scaling[36], as well as histology-derived assessments of cytoarchitecture[37,38] and cortical gene transcription[39–41] (for review see ref. [33]). Intriguingly, there is a strong correspondence between the relative positions of parcels along these gradients and the extent to which they share common cortical microstructure, connectivity, and profiles of gene expression, while the organization of cortical gradients differs between the two hemispheres[42–44]. Building upon these discoveries, a core goal of the present work is to characterize the organization and lateralization of the language network in relation to the mosaic of functionally distinct large-scale networks and associated macroscale connectivity gradients that span the cortical sheet.

Here, using a recently developed higher-order language atlas[8], we worked to determine the extent to which typical and atypical language lateralization is reflected across the functional architecture of the cerebral cortex. First, through a combination of resting-state functional MRI (fMRI) and task activation studies of language, we establish the presence of typical (92% of sample) and atypical (8% of sample) individuals within the Human Connectome Project (HCP) database[45]. Second, we provide evidence that atypical language lateralization is associated with global shifts in cortical organization. To do so, we applied the dimensionality reduction approach of diffusion map embedding[31] to resting-state data to extract a global framework that accounts for the dominant connectome-level connectivity patterns within each hemisphere. Individuals with atypical language organization exhibited corresponding hemispheric differences in the macroscale functional gradients. This pattern was preferential to functional

networks within association cortex. Third, twin-based heritability analyses revealed that both language lateralization and gradient asymmetries are, in part, driven by genetic factors. In doing so, our analyses reveal evidence linking the lateralization of language with broad changes in the functional organization of the cortical sheet.

## Results

### Identification of atypically lateralized individuals for language

We investigated the functional connectivity architecture of typically and atypically lateralized language functions in human cortex using task and resting-state fMRI data acquired at 3 T ($n = 995$, 110 left-handers) as a part of the Human Connectome Project[45]. Demographics are available in the *Methods* section (HCP participants). Language lateralization of each participant was assessed using SENSAAS, a higher-order language atlas[8]. In brief, two task-induced functional asymmetries during a language task[46] were obtained. First, at the network level, averaging across associated parcels, and second, within language network hubs, corresponding to Broca's and Wernicke's areas[8]. Three resting-state variables were also used to assess language lateralization. Two of which characterized the intra-hemispheric organization of the language network at rest, operationalized as the sum and the asymmetry of the average language network functional connectivity strength. The last metric characterized the homotopic inter-hemispheric connectivity of the language network. Taken together, these 5 functional metrics revealed the organization of the higher-order language network (Fig. 1A). The study sample was divided into groups based on their intra- and inter-hemispheric language network organization derived through an agglomerative hierarchical clustering procedure as described by Labache and colleagues[26].

Hierarchical classification established the presence of 3 groups: a strong typical group characterized by a strong leftward asymmetry during language task performance ($n = 480$, 36 left-handers), a mild typical group with moderate leftward asymmetry ($n = 433$, 48 left-handers), and atypical individuals showing a rightward asymmetry ($n = 82$, 26 left-handers, Fig. 1B), reflecting ~8 percent of the study population. The associated group demographics are available in Supplementary Table 1.

We next examined the extent to which language lateralization was reflected across each of the 5 features used to derive the higher-order language network (Fig. 1C). Analysis of covariance allowed us to replicate previously published results[26] conducted on an independent sample of 287 healthy volunteers from the BIL&GIN database[47]. Here, follow-up analyses were conducted to confirm that a single functional metric did not solely drive the results. Language lateralization was evident across each of the 5 functional language features included in the hierarchical classification (see Supplementary Tables 2–6). Task-induced functional asymmetries confirmed the rightward lateralization of atypical individuals both at the network ($\mu_{network} = -0.96 \pm 0.18$) and hubs level ($\mu_{hubs} = -1.16 \pm 0.26$, both corrected $p < 10^{-4}$), as well as a more leftward lateralized language network in strong ($\mu_{network} = 1.74 \pm 0.14$, $p_{network} < 10^{-4}$; $\mu_{hubs} = 2.64 \pm 0.19$, $p_{hubs} < 10^{-4}$) than in mild typical participants ($\mu_{network} = 0.70 \pm 0.12$, $p_{network} < 10^{-4}$; $\mu_{hubs} = 1.17 \pm 0.17$, $p_{hubs} < 10^{-4}$) both at the network and hub level. Intrinsic functional connectivity strength asymmetry profiles revealed that atypical individuals possess a bilateral language network organization ($\mu_{strength\ asym} = 9 \times 10^{-3} \pm 0.18$), in contrast to strong ($\mu_{strength\ asym} = 1.02 \pm 0.13$, $p_{strength\ asym} < 10^{-4}$) and mild typical ($\mu_{strength\ asym} = 0.85 \pm 0.12$, $p_{strength\ asym} < 10^{-4}$) participants. Strong and mild individuals showed no differences ($p_{strength\ asym} = 0.12$). Finally, strength sum and inter-hemispheric connectivity displayed a similar profile across groups, with strong typical ($\mu_{strength\ sum} = 12.16 \pm 0.39$ and $\mu_r = 0.61 \pm 0.02$) and atypical individuals ($\mu_{strength\ sum} = 12.06 \pm 0.53$ and $\mu_r = 0.61 \pm 0.03$, all $p > 0.93$) exhibiting a similar profile with significantly larger values than mild individuals ($\mu_{strength\ sum} = 9.40 \pm 0.35$ and $\mu_r = 0.49 \pm 0.02$, all $p < 10^{-4}$).

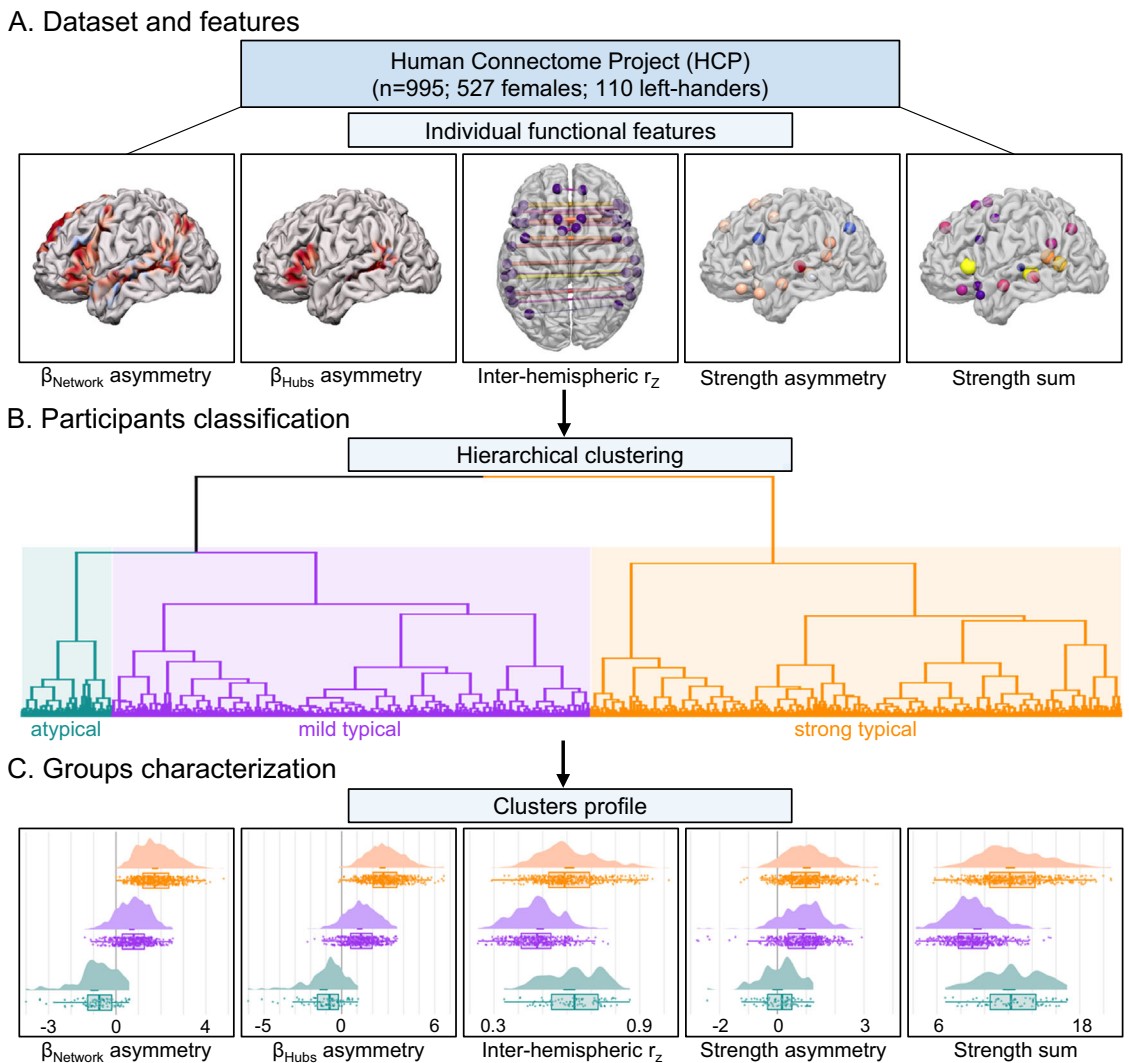

**Fig. 1 | Identification and characterization of language lateralization in 995 HCP participants.** Overview of preprocessing workflow. **A** The five individual functional metrics used to derive the sentence-processing supramodal network[26]. The average BOLD asymmetries values in the story-math contrast both at the network level ($\beta_{Network}$ asymmetry) and hubs level ($\beta_{Hubs}$ asymmetry), the average homotopic inter-hemispheric intrinsic correlation at the network level (inter-hemispheric $r_Z$), and both the asymmetry (strength asymmetry) and the sum (strength sum) of the average strength at the network level. **B** Hierarchical clustering resulted in the identification of three populations with varying degrees of language organization. Consistent with prior work[26], the first cluster with strong leftward asymmetries was named strong typical ($n = 480$, 36 left-handers, orange in the dendrogram), the second cluster exhibiting moderate leftward asymmetry was labeled mild typical ($n = 433$, 48 left-handers, purple in the dendrogram), and the third with strong rightward asymmetries was named atypical ($n = 82$, 26 left-handers, blue in the dendrogram). **(C)** Raincloud plots display the five functional metrics within each identified group ($n_{strong\ typical} = 480$, $n_{mild\ typical} = 433$, $n_{atypical} = 82$). $r_Z$, Fisher $z$-transformation correlation. Graphs display the density and boxplot (lower and upper hinges correspond to the 1st and 3rd quartiles, the middle line the median) of the five functional metrics values. Source data are provided as a Source Data file (see Data Availability).

## Gradient asymmetries and atypical lateralization

We next examined the extent to which the presence of typical and atypical language network lateralization may be evident throughout the functional organization of the cortical sheet. To do so, we took advantage of recent mathematical modeling of the functional topography of the cortex as proposed by Margulies and colleagues[31]. First, functional connectivity matrices ($384 \times 384$ AICHA parcels[48]) across the full sample were decomposed into components that capture the maximum variance in connectivity. Consistent with prior work[31,49], diffusion map embedding[50] was used to reduce the dimensionality of the connectivity data through the nonlinear projection of the voxels into an embedding space. The resulting functional components or manifolds, here termed gradients, are ordered by the variance they explain in the initial functional connectivity matrix. The present analysis focused on the first three gradients, reflecting divergent spatial patterns of connectivity across the cortex and accounting for 57% of the total variance in cortical connectivity. The first 3 group-level gradients respectively explained 22%, 21%, and 14% of the total variance in the initial matrix of cortical connectivity (Fig. 2).

In line with prior work[31,51–53], one end of the principal gradient of connectivity was anchored in unimodal (somato/motor and visual) regions, while the other end encompassed broad swaths of the association cortex, including aspects of the ventral and dorsal medial prefrontal, posteromedial/retrosplenial, and inferior parietal cortices, representing a functional hierarchy that spans from primary visual and somato/motor areas through the default network[34] (Fig. 2A), which underpins self-referential processing and core aspects of mental simulation[54–56]. Conversely, the second gradient peaked within unimodal networks, revealing a spectrum differentiating the somato/motor and auditory territories from the visual system (Fig. 2B). Lastly, the peak values in the third gradient (Fig. 2C) reflected a distinction between the frontoparietal network, spanning aspects of dorsolateral

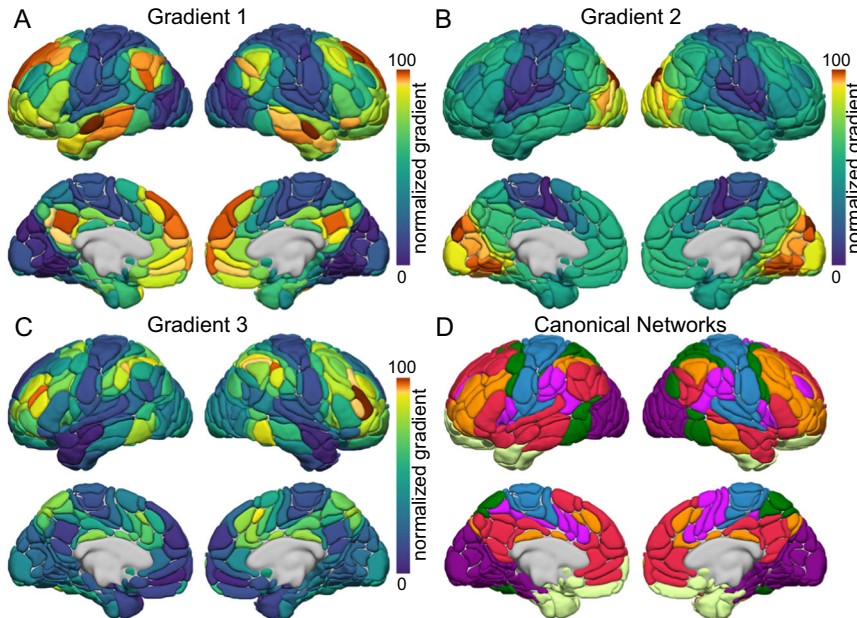

**Fig. 2 | Average dominant gradients of functional connectivity across the cortical sheet. The first three components resulting from diffusion embedding of the functional connectome connectivity matrix (as defined by Margulies and colleagues[31], dimension reduction technique = diffusion embedding, kernel = normalized angle, sparsity = 0.9). A** The principal gradient of connectivity transitioning from the unimodal (blue) to the association cortex (red). The proximity of colors reflects the similarity of connectivity patterns across cortex. The scale bar reflects z-transformed principal gradient values[42] derived from connectivity matrices using diffusion map embedding. **B** The second gradient primarily differentiates the somato/motor and auditory cortex (blue) from the visual system (red). **C** The third gradient reflects a network architecture contrasting frontoparietal (red) from default and somato/motor systems (blue). **D** Brain organization according to the 7 canonical networks identified in Yeo et al.[139] overlaid on the AICHA atlas parcels[48].

prefrontal, dorsomedial prefrontal, lateral parietal, and posterior temporal cortices, and the default network[57], placing the brain systems that underpin internally oriented cognition and those that coordinate responses to external task states and support complex cognition[58,59] along distinct ends of this organizational axis.

For each participant, functional network gradient asymmetry values correspond to the difference between the normalized gradient values in the left hemisphere minus the gradient values in the right hemisphere, averaged across all network parcels. Broadly, within the mild and strong typical groups, the first gradient showed a leftward asymmetry for 5 of the 7 canonical networks (average asymmetry values ranging from $\mu_{L\text{-}R(typical)} = 1.11$ to $\mu_{L\text{-}R(typical)} = 4.09$). The somato/motor network was symmetrical ($\mu_{L\text{-}R(typical)} = 0.01$, $CI_{95\%} = 0.24$). The control network was strongly right lateralized ($\mu_{L\text{-}R(typical)} = -6.21$, $CI_{95\%} = 0.59$) which may align with previous results on attention[60]. Conversely, the second gradient displayed a more heterogeneous pattern. Here, the gradient values within the control network ($\mu_{L\text{-}R(typical)} = -2.93$, $CI_{95\%} = 0.36$) were right lateralized, as well with the somato/motor ($\mu_{L\text{-}R(typical)} = -2.52$, $CI_{95\%} = 0.41$), limbic ($\mu_{L\text{-}R(typical)} = -0.63$, $CI_{95\%} = 0.48$), and default networks ($\mu_{L\text{-}R(typical)} = -0.54$, $CI_{95\%} = 0.26$). The visual network was symmetrical ($\mu_{L\text{-}R(typical)} = -0.24$, $CI_{95\%} = 0.29$), and the dorsal ($\mu_{L\text{-}R(typical)} = 0.88$, $CI_{95\%} = 0.54$) and ventral ($\mu_{L\text{-}R(typical)} = 1.26$, $CI_{95\%} = 0.45$) attention networks were left lateralized. Finally, the third gradient was primarily right lateralized or symmetrical, with the default ($\mu_{L\text{-}R(typical)} = -1.20$, $CI_{95\%} = 0.73$), limbic ($\mu_{L\text{-}R(typical)} = -2.53$, $CI_{95\%} = 0.71$) and salience/ventral attention networks ($\mu_{L\text{-}R(typical)} = -4.26$, $CI_{95\%} = 0.63$) rightward dominant, and the control ($\mu_{L\text{-}R(typical)} = -0.77$, $CI_{95\%} = 0.90$), dorsal attention ($\mu_{L\text{-}R(typical)l} = -0.61$, $CI_{95\%} = 0.67$) and somato/motor networks ($\mu_{L\text{-}R(typical)} = 0.34$, $CI_{95\%} = 0.34$) symmetrical. The visual network was the only leftward lateralized network for the third gradient ($\mu_{L\text{-}R(typical)} = 0.53$, $CI_{95\%} = 0.28$). See Supplementary Table 7 for a complete description of each network's typical gradient asymmetry values.

An important unanswered question is whether the broad and dissociable gradient asymmetries observed in individuals with typical and atypical language organization are uniformly distributed across the cortical sheet, or whether they are preferential to specific functional systems. Accordingly, we next tested the extent to which asymmetric profiles of network connectivity are evident within the atypical language participants. Here, mild and strong typical groups were merged into a single typical group[26] ($n = 913$, 84 left-handers) and next contrasted with the atypical participants ($n = 82$, 26 left-handers). Broadly, with exception of the limbic network, analyses of covariance revealed a preferential association between language lateralization and the asymmetric organization of association cortex networks, relative to unimodal systems across each of the three gradients (Fig. 3). These data suggest that the lateralization of language functions are carried throughout the '*association centres*' originally hypothesized by Paul Flechsig to underpin higher cortical functions and complex associative processing in humans[61]. See Supplementary Tables 8–10 for a full description of all the confound effects on gradient asymmetries, and Supplementary Table 7 alongside Supplementary Fig. 1 for a full description of each network gradient asymmetries values for both the typical and atypical groups.

Specifically, in the first gradient, five of the seven networks exhibited a significant main effect of language lateralization (all $p < 0.002$), of which 2 of them exhibited a shift in their lateralization from left to right dominant: the default network ($\mu_{L\text{-}R(atypical)} = -1.62$, $CI_{95\%} = 1.00$) and the salience/ventral-attention network ($\mu_{L\text{-}R(atypical)} = -1.65$, $CI_{95\%} = 1.05$). The dorsal-attention ($\mu_{L\text{-}R(atypical)} = 1.56$, $CI_{95\%} = 1.11$) and the visual network ($\mu_{L\text{-}R(atypical)} = 0.67$, $CI_{95\%} = 0.53$) showed a weakened dominance in the left hemisphere. The control network ($\mu_{L\text{-}R(atypical)} = -9.21$, $CI_{95\%} = 1.21$) was characterized by an increase in its dominance in favor of the right hemisphere. Those alterations in the hemispheric dominance mainly came from an increase of the gradient values in the right hemisphere for all the networks impacted by the phenotype, except the default network

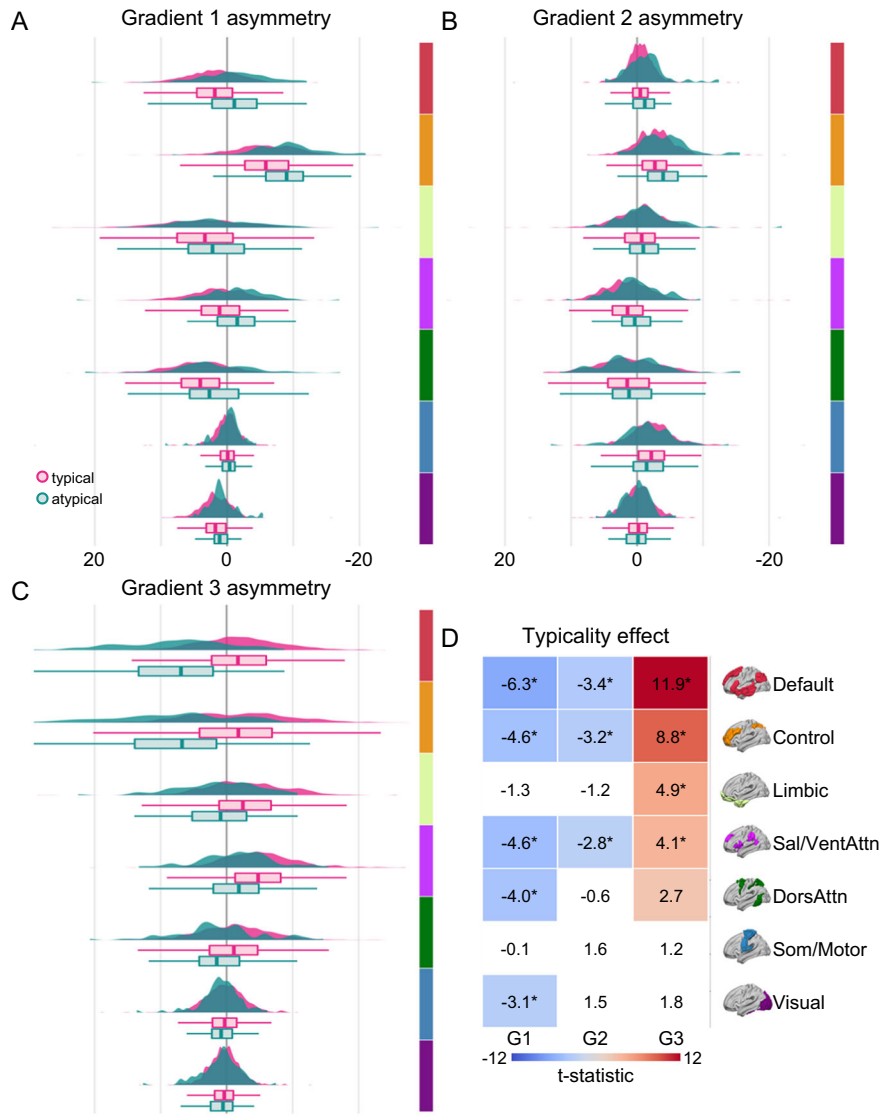

**Fig. 3 | Language lateralization is evident throughout the macroscale organization of the cortex. A–C** Network-level asymmetry (left *minus* right hemisphere) of the first three gradients for each language lateralization group. Colors reflect brain organization according to the 7 canonical networks[139] averaged across corresponding AICHA atlas parcels[48]. Graphs display the density and boxplot (lower and upper hinges correspond to the 1st and 3rd quartiles, the middle line the median) of individual gradient asymmetry values for the typical (magenta, $n = 913$) and atypical (teal, $n = 82$) groups. **D** The 2D grid displays the extent of language lateralization for each gradient and functional network. Values reflect the post-hoc *t*-statistic of the typicality main effect from ANCOVA. Post-hoc analyses were conducted using a two-sided Student's *t*-test. Colored cells display significant uncorrected effects ($p \leq 0.05$). Cells with a star are significant after Bonferroni correction for network number ($n = 7$, $p \leq 0.007$). In these analyses, each individual gradient has been scaled between 0 and 100. Source data are provided as a Source Data file (see Data Availability).

for which the change came from a decrease of the gradient values in the left hemisphere.

Three networks showed a significant impact of language lateralization within the second gradient (all $p < 0.006$), the default ($\mu_{\text{L-R(atypical)}} = -1.58$, $\text{CI}_{95\%} = 0.53$) and control networks ($\mu_{\text{L-R(atypical)}} = -4.26$, $\text{CI}_{95\%} = 0.74$) exhibited an increase in their rightward dominance, both coming from a mixed effect of an increase of their gradient values in the right hemisphere and a decrease in the left hemisphere. The salience/ventral-attention network ($\mu_{\text{L-R(atypical)}} = -0.17$, $\text{CI}_{95\%} = 0.92$) showed a symmetrical pattern in atypical individuals instead of being leftward dominant, coming from a decrease in its gradient values in the left hemisphere. Of the four remaining networks, two showed rightward lateralization: somato/motor ($\mu_{\text{L-R(atypical)}} = -1.28$, $\text{CI}_{95\%} = 1.28$), and limbic ($\mu_{\text{L-R(atypical)}} = -1.77$, $\text{CI}_{95\%} = 0.99$), and the two last ones were

symmetrical: dorsal-attention ($\mu_{\text{L-R(atypical)}} = 0.52$, $\text{CI}_{95\%} = 1.10$), and visual ($\mu_{\text{L-R(atypical)}} = 0.29$, $\text{CI}_{95\%} = 0.61$).

Finally, four of the seven networks for the third gradient significantly impacted by the language lateralization phenotype (all corrected $p < 10^{-4}$) showed an increase in their asymmetry in favor of the left hemisphere. The default ($\mu_{\text{L-R(atypical)}} = 9.05$, $\text{CI}_{95\%} = 1.52$), control ($\mu_{\text{L-R(atypical)}} = 8.51$, $\text{CI}_{95\%} = 1.87$), and limbic networks ($\mu_{\text{L-R(atypical)}} = 1.57$, $\text{CI}_{95\%} = 1.47$) became significantly left asymmetric through an increase of their gradient values in the left hemisphere and a decrease in the right. The salience/ventral-attention one showed a bilateralization ($\mu_{\text{L-R(atypical)}} = 1.25$, $\text{CI}_{95\%} = 1.31$) of its functional architecture coming from a decrease of its gradient values in the right hemisphere. The 3 remaining networks did not show significant differences compared to typical organization (all $p > 0.01$) and were all leftward symmetrical: dorsal-attention ($\mu_{\text{L-R(atypical)}} = 1.50$, $\text{CI}_{95\%} = 1.39$), somato/motor

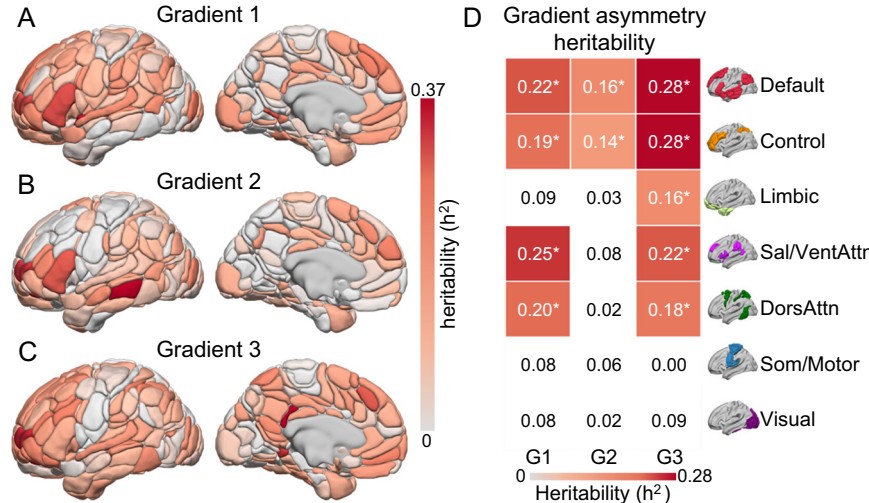

**Fig. 4 | The heritability of gradient asymmetry is evident in the macroscopic organization of the cortex. A–C** Regional-level heritability of the asymmetry (left minus right hemisphere) of the first three gradients. Warmer colors (solid red) indicate higher heritability of gradient asymmetry, for instance reflecting greater similarity among twins and siblings than unrelated individuals. **D** The 2D grid displays the extent of heritability of gradient asymmetry for each gradient and functional network. Values reflect the heritability of gradient asymmetry. Heritability of network asymmetry was estimated across 7 canonical functional networks using SOLAR[70] ($n = 989$: 130 MZ twins, 70 DZ twins, 479 non-twin siblings, and 110 unrelated singletons). Colored cells with a star reflect networks with significant heritability of gradient laterality after Bonferroni correction ($n = 7$, $p \le 0.007$ corrected).

$(\mu_{L-R(atypical)} = 0.84$, $CI_{95\%} = 0.71)$, and visual $(\mu_{L-R(atypical)} = 1.12$, $CI_{95\%} = 0.58)$.

### Heritability of language lateralization and gradient asymmetry

Population-based neuroimaging studies have revealed the influence of genetic factors on the connectivity strength[62], size, and spatial organization[63] of large-scale cortical networks. However, although prior work has begun to catalog the evolution[64], development[49,65], and organization[66,67] of the brain's functional architecture, the role of genetics in sculpting the lateralization of cognitive functions and associated asymmetries in the macroscale organization of cortex has remained largely unexplored. Prior work has established the heritability of gradient asymmetries in humans, an aspect of cortical organization likely present across non-human primates[44]. Other studies have highlighted the association between differences in structural asymmetry in left- and right-handed people and the genetic basis of their manual preference[68], as well as the genetic basis of structural asymmetries[69]. Here, to advance our understanding of the biological bases of hemispheric specialization, we worked to determine heritable sources of variation that may govern the lateralization of both localized language functions and functional gradient asymmetries across the cortical sheet[69].

Leveraging a twin-based estimate of heritability, our analyses suggest that both the lateralization of the language network, as assessed through the hierarchical classification approach ($h^2 = 11.2\%$, SE = 6%, $p = 0.038$), and the hemispheric asymmetries in gradient organization, reflecting the difference in gradient values between the left and right hemisphere (Supplementary Table 11, G1: 14.4%, SE = 6%, $p = 0.007$; G2: 2.0%, SE = 5%, $p = 0.36$; G3: 24.0%, SE = 6%, $p < 10^{-4}$), are under genetic control. Heritability of gradient asymmetry values for each network was estimated using sequential oligogenic linkage analysis routines (SOLAR[70]) and covaried for age, sex, age$^2$, age × sex, age$^2$ × sex, handedness, and FreeSurfer-derived intracranial volume. Prior work examining connectivity strengths and the network topographies indicates reduced heritability in the size[63] and connectivity strength[62] of heteromodal association networks, relative to unimodal sensory/motor cortex. In contrast, with the exception of the limbic network, the present analyses revealed the influence of genetic factors on the gradient asymmetries across each association cortex network.

Notably, genetic factors did not significantly account for the lateralization of gradient values within somato/motor and visual territories and heritability was significantly greater (Fig. 4; $p < 10^{-3}$) within heteromodal ($h^2$ μ = 18.5%, SD = 7.7%) association cortices than within unimodal networks ($h^2$ μ = 5.5%, SD = 3.8%). Overall, these data reveal the substantial influence of genetic factors on the lateralization of both specific cognitive functions and the broad functional organization of the cortex. These results are consistent with the hypothesis that neuronal asymmetries likely developed under phylogenetic pressure, and therefore possess a genetic basis[71].

### Discussion

Our present study reveals that asymmetrical language network organization is broadly reflected throughout the global connectivity structure of the cerebral cortex. Using task-evoked and resting-state data, we identify a pattern of atypical language network lateralization and corresponding alterations in functional coupling across the cortical sheet in ~8 percent of individuals. These group-level changes in connectivity are preferential to transmodal association cortex and heritable, providing evidence that both the lateralization of the cortical territories supporting language and the associated functional processing streams they are embedded within are under genetic control. Together, this work advances our understanding of the relationship between the localized hemispheric specialization of specific behaviors and the hierarchical functional axes that capture the topographic organization of large-scale cortical networks.

Hemispheric specialization reflects a core property of human cognition and a marker of successful development. In most individuals, the left hemisphere is specialized for language and motor control of their dominant hand, whereas the right hemisphere plays a preferential role in visuospatial processing[72,73]. Across development, the maturation of lateralized functions is associated with improved visuospatial and language abilities and enhanced cognitive efficiency[74,75]. Although the precise timing of these developmental cascades remains largely unknown[14,16], some markers of lateralization are already apparent during gestation, for instance the leftward asymmetrical folding of the Sylvian fissure[76]. At birth, intra-hemispheric white matter connectivity across the language network is reduced in favor of strong inter-hemispheric coordination[77], an

organizational property that is consistent with the presence of prominent homotopic intrinsic connectivity from 3 to 6 months of life[78]. Suggesting a complex developmental picture that is sensitive to task state, when newborns are presented with spoken words they exhibit a left-lateralized asymmetric response, whereas brain connectivity at rest remains evenly distributed across hemispheres[79]. Although multiple factors appear to contribute to the emergence of lateralized functions[80], atypical language organization and more globally atypical functional lateralization are preferentially evident in individuals with neurodevelopmental, psychiatric, and/or neurological disorders[11,22,81], such as autism[82–85] and schizophrenia[86,87]. Here, in a population of healthy young adults, we observed atypical lateralization in a sizable minority of the study sample. Critically, although individuals with language and/or cognitive impairments often present with altered cerebral lateralization, the vast majority of people with atypical language lateralization have no corresponding cognitive deficits[88,89], or exhibit slightly lower visuospatial and verbal memory performance compared to strong leftward lateralized individuals[26]. As such, understanding the interactions linking the biological underpinnings of language lateralization, cognition, and illness risk would have significant implications for both developmental biology and cognitive neuroscience[22].

The cerebral cortex is composed of areal parcels, embedded within a set of distributed large-scale networks, and positioned within corresponding processing streams[1,33,34,90]. Cortical networks exhibit a parallel and tightly interdigitated organizational structure. As a result, the language system may impinge upon, and be influenced by, putatively distinct yet spatially adjacent networks[91]. Extending upon recent evidence indicating that regional anatomical asymmetries are reflected throughout the entire brain[92,93], our analyses revealed a relationship between the lateralization of language functions and the sweeping functional gradients that capture the topography of large-scale networks across cortex. These results are consistent with mounting evidence suggesting an association between language lateralization and asymmetry in functional connectivity at rest[94,95], with local brain efficiency in atypical individuals[96], and with the presence of individual-specific functional deviations, or "network variants," that systematically differ across hemispheres[97]. Of note, across all three gradients the observed differences between the typical and atypical groups were preferential to association networks. Over the course of vertebrate evolution, the evolutionary enlargement of the cortical mantle in primates has been preferentially localized within spatially distributed aspects of association cortex, relative to the primary and secondary sensory systems, perhaps allowing for the development of novel capabilities independent from the primary senses[64,98]. The present profile of results is consistent with literature indicating hemispheric asymmetries for both language and attentional allocation as well as theories suggesting that phylogenetically expanded aspects of cortex, for example inferior parietal lobule, reflect the cortical territories with the most prominent structural and functional asymmetries in humans[99]. Critically, our analyses cannot establish the specific biological cascades that influence the emergence of atypical language laterality or corresponding shifts in the broad functional architecture of cortex. Rather, these data highlight the importance of future work to identify underlying processes that contribute to the development of functional asymmetries throughout the cortical sheet and associated hemispheric differences in information processing. Across each of the three main gradients, the lateralization of language appeared to preferentially reverberate throughout the functional architecture of association cortex. Although speculative, the present analyses suggest that the lateralization of isolated functions, such as language, may be tightly tied to the lateralization of a host of other seemingly independent processes.

The genetic origin of language capacities[100], and other properties of hemispheric specialization reflect a fundamental question in cognitive neuroscience with clear relevance for the study of both health and disease[12]. Prior work indicates that intrinsic connectivity between language related regions[101], as well as evoked brain activations during language tasks[102], are heritable. Our present analyses indicate a clear genetic basis for population-level patterns of language lateralization and corresponding features of cortical organization. These results are consistent with recent work highlighting that the genetic contributions to variation in handedness are complex and polygenic[103,104], suggesting diverse biological pathways may converge in the atypical phenotype[105,106]. Here, the heritability of gradient asymmetries were evident across the cortical sheet, preferential to functional networks within association territories. Prior work has revealed core principles that govern the evolution, development, and organization of large-scale brain networks. Broadly, in contrast to our present analyses, these data have suggested relaxed genetic control of association cortices relative to primary sensory/motor regions[63]. A profile of heritability that is consistent with the presence of increased population-level variability in functional connectivity[107], relative network sizes, and topographic network similarity in association relative to unimodal cortex[63,108–110]. The discovery of the increased influence of genetic factors on the gradient asymmetries across each association cortex network raises the possibility that, despite the broadly reduced heritability of association cortex functions, features of brain lateralization remain preferentially influenced by genetics. One speculation is that our present results reflect two partially distinct developmental paths, first an initial genetically mediated developmental cascade biasing fundamental aspects of brain lateralization. Second, once the genetically mediated plan is laid out, the subsequent protracted development of association cortex functions provides for a period of prolonged plasticity and increased sensitivity to environmental inputs[111,112].

Although the present results provide evidence that atypical language network organization is linked to widespread properties of cortical organization, there are several limitations with our approach. First, the causal pathways linking the lateralization of the putatively distinct cognitive processes with broad features of brain functioning remain to be established. We are unable, for instance, to examine atypicalities in the lateralization of attentional processes. Additionally, given the cross sectional nature of the study data, we are not able to establish the developmental course linking atypicality in language lateralization with the development of global brain architecture. As such, the manner through which lateralized functions impinge upon, and are in turn influenced by, other properties of brain organization across development remains an open question. From a methodological perspective, the HCP database contains only one language comprehension task[46]. Of note, brain responses to language comprehension have been shown to share consistent lateralization patterns with other language tasks[113]. Additionally, although an auditory task exhibits less lateralization than a production task (which is the most lateralized task) or a reading task, it still significantly activates and lateralizes the language network in the left hemisphere[8]. In line with this, asymmetries observed during language tasks, compared to a high reference condition, serve as a suitable marker for determining language dominance[114]. Furthermore, incorporating asymmetries at the hubs (Broca's and Wernicke's areas) increases the amplitude of asymmetries in the comprehension task. Finally, our prior work showed that combining resting and task metrics is essential to accurately identify atypical individuals[26]. Additionally, from a clinical perspective, the potential relationships linking aphasias with corresponding shifts in the global functional architecture of cortex have yet to be established and warrant further study[115,116].

The extent to which the lateralization of specific cognitive functions may be evident across the macroscale organizational properties of the cortical sheet is a central question across the brain sciences. The present results demonstrate that asymmetric language network organization is carried throughout the association cortex. While the exact

determinants of lateralization mechanisms are still unknown, both the hemispheric specialization of language and corresponding asymmetries across the sweeping functional gradients that span cerebral cortex were found to be heritable. Here, the lateralization of heteromodal association cortex networks under increased genetic control, relative unimodal networks. The further study of this entangled relationship between language lateralization and broader properties of functional network organization has the potential to shed light on the phenomenon of cerebral dominance thought to underpin sophisticated cognition in humans[117] as well as neuropsychiatric and neurological disorders with known alterations in brain laterality[118,119].

## Methods
### HCP participants
The study sample was part of the S1200 Release (updated April 2018) of the WU-Minn Human Connectome Project (HCP) database that has been fully described elsewhere[45]. From 1206 healthy participants, participants with fully completed 3 T language and 3 T resting-state fMRI protocols were selected, resulting in a total of 995 participants (477 women). The mean educational level of participants was 14.97 years (SD = 1.77 years). The sample mean age was 28.70 years (SD = 3.71 years). Participants' handedness was defined based on the manual preference strength assessed with the Edinburgh inventory:[120] participants with a score below 30 were considered left-handers[121,122], right-handers otherwise. The values for Edinburg Inventory (EHI) scores can be found in the HCP database's restricted demographic file under the variable name "*Handedness*", which is accessible only to authorized users. It is worth noting that the HCP's EHI score includes one foot-related item among its ten items. As a result, it does not solely measure manual preference. Analyses using a corrected EHI score[123] that excludes the foot-related item yield consistent results (see supplementary material: *Replication of results using a different Edinburgh score* section). The sample contained 110 left-handed participants (50 women), leading to a sample broadly representative of the general population[122,124]. Data collection was approved by a consortium of institutions institutional review boards (IRBs) in the United States and Europe, led by Washington University (St Louis) and the University of Minnesota (WU-Minn HCP Consortium). The current study was approved by the Yale University IRB.

### MRI data preprocessing
HCP datasets used include two main imaging sessions. Data were acquired using multiband echo-planar imaging (EPI) on a customized Siemens 3 T MRI scanner (Skyra system). Structural data consisted of one 0.7 mm isotropic scan. (1) Two sessions (REST1 and REST2) of resting-state fMRI (rs-fMRI), where each session comprised two runs (left-to-right, and right-to-left, phase encoding) of 14 min and 33 s each (repetition time (TR) = 720 ms, echo time (TE) = 33.1 ms, voxel dimension: 2 mm isotropic). Details on rs-fMRI can be found elsewhere[125]. (2) Task fMRI (t-fMRI) data were acquired using the identical multiband EPI sequence as the rs-fMRI session. Among the 7 contrasts, only the Story *minus* Math contrast was used. Details on the protocol are available elsewhere[46]. The language-related protocol was developed by Binder and colleagues[126]. Briefly, the contrast consisted of comparing comprehension of brief narratives (Story task) with a semantically shallow control task involving serial arithmetic (Math task). Two runs were performed each consisting of 4 blocks of a Story interleaved with 4 blocks of a Math task. Each run was 3.8 min long.

Minimally preprocessed volumetric rs- and t-fMRI data were sourced from the online HCP repository through Amazon Web Services (AWS). Details of the minimal preprocessing pipeline can be found elsewhere[127]. The R library *neurohcp*[128] was used to interface AWS S3 bucket (R package version: 0.9.0). The R library *RNifti*[129] (R package version: 1.3.1) and *oro.nifti*[130] (R package version: 0.11.4) were used to read and handle the fMRI data.

The 995 individuals have been coregistered using MSM-All pipeline. t-fMRI data are represented in the HCP 32k_LR MNI surface space[131], since volume-smoothed level 2 t-fMRI analysis results are no longer being distributed. rs-fMRI data are represented in the MNI volumetric space.

### Language atlas statistics
Preprocessed data were analyzed to compute 5 functional features characterizing the high-order language network. These 5 features have been previously shown to accurately determine the language network typicality[26].

The high-order language atlas (SENSAAS) has been fully described elsewhere[8]. Briefly, 18 regions of interest corresponding to the core language network have been selected from the language atlas. The core language network corresponded to a set of heteromodal brain regions significantly involved, leftward asymmetrical across 3 language contrasts (listening to, reading, and producing sentences), and intrinsically connected. It should be noted that the language atlas was based on the AICHA atlas, a functional brain atlas optimized for the study of functional brain asymmetries[48].

First, two of the 5 features were computed from the t-fMRI data. For each individual, the native volumetric language atlas has been mapped to the closest mid-thickness surface vertex using tools from the HCP workbench[132]. The surface language atlas was then used as a binary mask to estimate the average BOLD signal variation of language networks in both hemispheres for the Story *minus* Math contrast. The average asymmetry of activations was then measured by computing the difference between the left and right hemispheres (left-right). The same process has been repeated to estimate the average asymmetry at the hub level. A description of language hubs can be found in Labache and colleagues[8]. Briefly, the language network hubs corresponded to the inferior frontal gyrus (Broca's area) and to the posterior aspect of the superior temporal sulcus (corresponding to Wernicke's area).

Second, the 3 other features were computed from the rs-fMRI data. For each of 4 rs-fMRI scans, each individual and each of 18 language regions, an individual BOLD rs-fMRI time series was computed by averaging the BOLD fMRI time series of all voxels located within the region's volume. An intrinsic connectivity matrix was then calculated for each of 995 individuals and scans. The intrinsic connectivity matrix off-diagonal elements were the Pearson correlation coefficients between the rs-fMRI time series of region pairs. For each individual, the 4 connectivity matrices were z-transformed prior to being averaged and r-transformed with a hyperbolic tangent function. The 4 scans were averaged to increase the signal-to-noise ratio and reliability for generating individual functional connectivity matrices[133]. For each individual and each region, the strength, or centrality degree, was computed in each hemisphere. The strength was calculated as the sum of the correlations existing between one region and all the 18 others. Strength values were then averaged across the 18 regions of the same hemisphere and the resulting left and right averaged strength values were summed. The left minus right differences were also computed. Finally, the inter-hemispheric connectivity strength was estimated in each individual by averaging across the 18 region pairs of the z-transformed intrinsic correlation coefficient between homotopic regions.

### Connectivity embedding
For each participant, values were obtained for the first 3 functional gradients. The gradients reflect participant connectivity matrices, reduced in their dimensionality through the approach of Margulies and colleagues[31]. Functional gradients reflect the topographical organization of cortex in terms of sensory integration flow as described by Mesulam[134]. Gradients were computed using Python[135] (Python version: 3.8.10) and the *BrainSpace* library[136] (Python library version: 0.1.3).

Gradients computed at both the regional and vertex level showed similar performance[136].

Average region-level functional connectivity matrices of the 4 scans were generated for each individual across the entire cortex (*i.e.* 384 AICHA brain regions, same process as for the language connectivity matrices). Consistent with prior work, the top 10% connections of each region were retained, and other elements in the matrix were set to 0 to enforce sparsity[31,49]. The normalized angle distance between any two rows of a matrix was calculated to obtain a symmetrical similarity matrix. Diffusion map embedding[50,137,138] was implemented on the similarity matrix to derive the first 3 gradients. Note that the individual-level gradients were aligned using Procrustes rotation ($N_{iterations}=10$) to the corresponding group-level gradient. This alignment procedure was used to improve the similarity of the individual-level gradients to those from the prior literature. Min-max normalization (0-100) was performed at the individual level for the whole brain[42].

To keep the subsequent analysis circumscribed to large-scale network brain organization, gradients values have been averaged, for each participant, according to each of the 7 canonical networks described by Yeo and colleagues[139]. Prior to the averaging step, each AICHA region has been assigned to one of the 7 canonical networks based on its spatial overlap with a given network. Gradient asymmetry was then computed for each participant and region. For a given network, gradient asymmetry corresponded to the difference between the normalized gradient value in the left hemisphere minus the gradient values in the right hemisphere.

### Statistical Analyses

Statistical analysis was performed using R[140] (R version: 4.1.0). Data wrangling was performed using the R library *dplyr*[141] (R package version: 1.0.10). Brain visualizations were realized using Surf Ice[142].

An overview of our experimental workflow is shown in Supplementary Fig. 2.

**Language lateralization identification**. Using the same methodology as by Labache and colleagues[26], the 995 participants have been classified using agglomerative hierarchical clustering. Each participant was characterized according to their language network organization. Language network was described by 5 features: network- and hubs-level asymmetry during the language task, sum and asymmetry of strength, and homotopic inter-hemispheric connectivity value at rest. Hierarchical classification allowed for the identification of language lateralization for each individual. Briefly, hierarchical agglomerative clustering[143] was performed using Euclidean distance as metric and Ward's criterion as linkage criteria[144]. Each variable was standardized before classification.

We employed an unsupervised methodology to determine the optimal number of clusters. Combining results from the R package *NbClust*[145] (R package version: 3.0.1) and *pvclust*[146] (R package version: 2.2.0), we selected a 3-cluster solution (strong typical, mild typical, and atypical), which was shown to reproduce our previous results[26]. *pvclust* showed that the 3-cluster solution was stable (Approximately Unbiased *p*-value = 0.98, $CI_{95\%} = \pm 0.002$). Furthermore, among the 26 indices used by *NbClust* to evaluate the stability of different clustering schemes, the 3-cluster partition was the second most supported, with 5 indices in its favor. The most supported partition was a 4-cluster solution with 11 indices. This would have led to the partitioning of the strong typical group, which was deemed unnecessary as it would not have provided new insights to the aim of this study. The 3 clusters defined the language lateralization phenotype.

Using analysis of covariance, the broader relationship between language lateralization and the 5 features was assessed. Each of the 5 models was specified as follows: the features were the dependant variable, language lateralization phenotype was the independent variable, age, intracranial volume (FreeSurfer-derived), gender, and handedness as covariate, as well as the interaction handedness × language lateralization phenotype. Post-hoc analyses were conducted using Tukey's range test for multiple comparisons (to account for the number of language lateralization phenotype: strong typical, mild typical, and atypical), or Student's *t*-test for binary ones. The reported *p*-values in the corresponding *Results* section are corrected for multiple comparisons.

**Language Lateralization impact on gradient asymmetry**. Analysis of covariance was used to assess the broader relationship between language lateralization and lateralization of large-scale cortical organization, modeled by gradient asymmetry. For each gradient and network, the model was specified as follows: gradient asymmetry for a given network and a given gradient was the dependant variable, language lateralization phenotype was the independent variable, age, intracranial volume (FreeSurfer-derived), gender, and handedness as covariate, as well as the interaction handedness × language lateralization phenotype. Bonferroni correction of significance thresholds was used to account for 7 independent tests of a given gradient. Post-hoc analyses were conducted using the Student's *t*-test. The reported *p*-values in the corresponding *Results* section are not corrected for multiple comparisons and are significant if less than 0.007.

**Heritability of Gradient Asymmetry and Language Lateralization Phenotype**. Heritability is a statistic indicating to what extent the variation in a phenotypic trait is accounted for by the combined effects of genetic variations over the genome of a population. Heritability estimates range from 0 to 1. The heritability of gradient network asymmetry was estimated using SOLAR[70] (version: 9.0.0) through the R package *solarius*[147] (R package version: 0.3.2), covarying for age, sex, $age^2$, age × sex, $age^2$ × sex, handedness, and FreeSurfer-derived intracranial volume. Bonferroni correction of significance thresholds was used to account for 7 independent tests of heritability. Heritability of language lateralization phenotype (*i.e.* being typical or atypical) was also assessed using the same covariates.

Heritability estimates were conducted on 989 HCP participants, composed of 130 MZ twins ($n = 260$), 70 DZ twins ($n = 140$), non-twin siblings ($n = 479$), and unrelated singletons ($n = 110$).

### Reporting summary

Further information on research design is available in the Nature Portfolio Reporting Summary linked to this article.

## Data availability

This study used publicly available data from the HCP (https://www.humanconnectome.org/). Data can be accessed via data use agreements. The language atlas is available here: https://github.com/loiclabache/SENSAAS_brainAtlas. Source data are provided with this paper and are also available here: https://github.com/loiclabache/Labache_2022_AO/tree/main/Data. Source data are provided with this paper.

## Code availability

The code used in the Method section to process the data from the HCP and reproduce the results and visualizations can be found here:[148] https://github.com/loiclabache/Labache_2022_AO.

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

## Acknowledgements

This work was supported by the National Institute of Mental Health (R01MH120080 and R01MH123245 to AJH) as well as by the following awards to BTTY: the Singapore National Research Foundation (NRF) Fellowship (Class of 2017), the NUS Yong Loo Lin School of Medicine (NUHSRO/2020/124/TMR/LOA), the Singapore National Medical Research Council (NMRC) LCG (OFLCG19May-0035), the NMRC STaR (STaR20nov-0003), and the Singapore Ministry of Health (MOH) Centre Grant (CG21APR1009). Any opinions, findings and conclusions or recommendations expressed in this material are those of the authors and do not reflect the views of the Singapore NRF, NMRC or MOH. Data were in part provided by the Human Connectome Project, WU-Minn Consortium (Principal Investigators: David Van Essen and Kamil Ugurbil; 1U54MH091657) funded by the 16 NIH Institutes and Centers that support the NIH Blueprint for Neuroscience Research; and by the McDonnell Center for Systems Neuroscience at Washington University.

## Author contributions

L.L. and, A.J.H. designed the research. L.L. conducted the research. L.L., A.J.H, T.G., and B.T.T.Y. analyzed and interpreted the results. L.L., and A.J.H. wrote the paper, which all authors commented on and edited. L.L. and A.J.H. made figures. L.L. analyzed the data. L.L. published the code. All authors provided analytic support.

## Competing interests

The authors declare no competing interests.
