## [Peer Review File · Nature Communications]

Language network lateralization is reflected throughout the macroscale functional organization of cortexReviewer #1 (Remarks to the Author):

The authors provide an interesting analysis of the Human Connectome Project (HCP) data, exploring the relationship hemispheric asymmetry for language processing and functional connectivity/gradient asymmetries, as well as utilizing the twin/family data provided by HCP to explore heritability of these features. The chosen approach is well-founded in the literature and the findings contribute to a better understanding of hemispheric functional asymmetry, one of the core features of human brain organization. Thus, I personally welcome this research and would love to see it published in NComms. However, the manuscript in its current form raises two major and one minor concern, which I feel deserve clarification before considering publication.

Firstly, the authors write that handedness was assessed with the Edinburgh Inventory (EHI) and that participants with a score below 30 were classified as left-handers (Methods, first section). As handedness was used as covariate in all analyses (and not just as descriptive information), I feel it would be beneficial to get a bit more information on the handedness classification applied by the authors. Firstly, given that no additional information is provided, I assume that the variable "Handedness" from the restricted HCP data was used. This variable is, however, problematic. That is, the HCP testing included only 9 of the 10 hand preference questions (leaving out "drawing") from the EHI but included the "footedness" question (see e.g. Ruck & Schoenemann, 2020, *Laterality*: DOI: 10.1080/1357650X.2020.1866001) and it appears that the HCP variable Handedness is calculated by including "footedness" as the 10th item (at least that is what this preprint claims: doi: 10.1101/2022.11.14.516402). Why this is the case, is not clear, but it likely is an "oversight" in HCP data handling. But as handedness and footedness are far from perfectly correlated (see doi: 10.1038/s41598-020-71478-w), "HCP Handedness" cannot be used to determine hand preference without leaving doubts. Thus, if the authors indeed have used the HCP handedness variable, this requires (and I hate writing this) reanalysis of the data with a corrected handedness variable. Alternatively, if the authors were aware of the problem and retrieved the item-wise EHI answers from HCP and calculated an "own" laterality score, this information needs to be provided to the reader. In this case, it would also be interesting to hear how this was done, and how the laterality score was scaled. And why "30" was considered the correct cut-off for separating left and right handers. Considering the standard range from -100 to 100 suggested by the EHI, "30" appears somewhat arbitrary as a cut-off as it would actually include individuals with a slight right hand preference in the left hander sample.

Secondly, if I understand the methods description correctly, 3 of the 5 features that are used for the asymmetry phenotype classification are derived from the HCP rs-fMRI, while the same rs-fMRI was utilized to derive the 3 gradient asymmetries. That means that the rs-fMRI data contributed (although with different processing steps and to different degree) both to determining the independent and dependent variable used in the second set of analyses (presented under "Gradients asymmetries and atypical lateralization"). Consequently, it is not clear whether the found "gradient" differences between the phenotype groups, simply reflect that the groups were determined (partly) by the same data. Thus, the authors need to convince that this is not the case.

Finally, I feel the manuscript is densely written and could benefit from thorough language editing. Many sentences are (at least in my opinion) unnecessarily complicated and make it difficult to understand what the authors mean. One of several examples: "The parallel and interdigitated organization of cortical networks suggests that the language system may impinge upon, and be influenced by, putatively distinct yet spatially adjacent networks" (top of 2nd page of discussion). As nice as this sounds, does this not just say that language processing relies on cortical networks that may interact (which is not surprising).

Reviewer #2 (Remarks to the Author):

The study uses data from the Human Connectome Project (HCP) to quantify hemispheric language dominance based on task fMRI, and relate it to hemispheric differences in macroscale functional gradients based on resting fMRI. In addition, the study shows heritability of language laterality and

gradient asymmetries using HCP twin data. The findings contribute to understanding how specific task-functional organization can relate to broader aspects of brain network organization. The findings might also provide a route to using large-scale resting fMRI data to perform molecular genetic studies of hemispheric language dominance. I find the study clearly described and a useful contribution to the field. Further consideration could be given to the following issues:

- The variation across individuals is quantitative and continuous, but the authors rely on clustering to create groups (categories) for subsequent analyses. The particular clusters that were defined do not appear especially robust compared to alternative cluster solutions. The issue could be avoided by treating the data as continuous. If not, then a stronger rationale could be given for the cluster-based approach, together with clearer indications that the chosen solution was robust compared to others.

- Hemispheric language dominance is most pronounced for language production tasks. The particular task contrast used in HCP involves comparing comprehension of brief narratives with an arithmetic task. The Discussion would benefit from some consideration of this issue. Was the HCP task optimal for determining hemispheric language dominance?

- The study reports heritability of lateralization of the language network, and also heritability of hemispheric asymmetries in gradient organization. The same data and software could be used to assess the genetic correlation between these two types of measure, i.e. to what extent do the same genetic variants affect hemispheric language dominance and asymmetries of gradient organization, versus being independently heritable (and therefore correlated for non-genetic reasons).

- For introducing/discussing the genetic parts it would be relevant to cite recent genetic association studies of left-handedness and brain asymmetry:

<https://www.nature.com/articles/s41562-020-00956-y>

<https://www.nature.com/articles/s41562-021-01069-w>

<https://www.pnas.org/doi/10.1073/pnas.2113095118>

<https://www.nature.com/articles/s41598-019-42515-0>

<https://academic.oup.com/brain/article/142/10/2938/5556832>

Reviewer #3 (Remarks to the Author):

Leveraging a higher-order language atlas and following previous work, the authors performed a hierarchical clustering based on a combination of resting-state and task-evoked fMRI features. They identified 3 subgroups of subjects related to language lateralization in the HCP database. Gradients were computed through diffusion map embedding on functional connectivities among the AICHA atlas ROIs. Asymmetry of each gradient in typical subjects was calculated as the difference between gradient values in the left and right hemispheres for each canonical network.

Then, the impact of language lateralization on the asymmetry of each canonical network was assessed using an ANCOVA, comparing atypical subjects for language lateralization and typical and mild typical subjects merged into a single group. Finally, using a multidimensional heritability analysis, genetic contribution to the phenotypic variance of language network lateralization, on the one hand, and the hemispheric asymmetries in gradient organization, on the other hand, was assessed. In addition, the heritability of gradient asymmetry values for each canonical network was estimated.

The manuscript is well-written, and the subject is highly interesting and original. Many analyses have been carried out, using various methods and uncovering valuable results. However, I would have some questions about several points.

The experimental design of this work is very complex, and for the sake of clarity, the author could provide a diagram of the relationships between the different analyses and results.

When identifying subgroups of subjects based on their language network lateralization using hierarchical clustering, the number of subgroups was set to 3, based on previous work (Labache et al. *Elife* 2020). In this previous work, the cohort was enriched for left-handed subjects with no twin pairs or siblings, which is not the case in HCP. Moreover, input features based on task-fMRI

data were derived from production, reading and listening tasks versus a Story-Math contrast in the current study. I wonder if the 3-subgroup result in Labache et al. Elife 2020 could be directly applied here, without any machine learning optimization of this hyperparameter, considering that the input is different? It would be valuable to reproduce this result in similar but not identical conditions.

Multiple test correction is not mentioned in the ANCOVA of language lateralization and the 5 input features of hierarchical clustering. Are the reported p-values corrected or not? In supplementary tables 10 and 11, when correcting for multiple testing, the correction accounted for 7 networks, but there were also 3 gradients for each network; I wonder if the p-values should have been corrected for 21 tests instead of only 7, as in supplementary figure 2? Same question for ANCOVA of language lateralization and lateralization of large-scale cortical organization.

The authors conclude that genetic factors substantially influence the lateralization of both specific cognitive functions (language, I suppose) and the broad functional organization of the cortex. As covariance between language lateralization and canonical networks lateralization was assessed, the genetic correlation between these could also be computed to give even more insight into the genetic architecture of brain lateralization.

The supplementary figure 2 is never commented on.

For clarity, the third paragraph of "Gradients asymmetries atypical lateralization" should refer to Supplementary table 10. Typo at the end of paragraph : dorsal-attentional ($\mu_{\text{typical}}=-0.61\dots$)
→ dorsal-attentional ($\mu_{\text{L-R(typical)}}=-0.61\dots$)

I would recommend this work for publication in Nature Communications with major revisions.

RESPONSE TO REVIEWS

We are grateful to the reviewers for their close read of our manuscript and are highly pleased by their enthusiasm. The reviewers raised important clarifying points and suggested additional analyses. We have carefully considered and addressed these comments and believe the manuscript is greatly strengthened. Detailed responses to each comment are included below. Reviewer comments are shown in *italicized font*, answers are shown in black, and new text is shown in blue.

REVIEWER 1

The authors provide an interesting analysis of the Human Connectome Project (HCP) data, exploring the relationship hemispheric asymmetry for language processing and functional connectivity/gradient asymmetries, as well as utilizing the twin/family data provided by HCP to explore heritability of these features. The chosen approach is well-founded in the literature and the findings contribute to a better understanding of hemispheric functional asymmetry, one of the core features of human brain organization. Thus, I personally welcome this research and would love to see it published in NComms. However, the manuscript in its current form raises two major and one minor concern, which I feel deserve clarification before considering publication.

We thank the reviewer for the positive evaluation of our work.

Firstly, the authors write that handedness was assessed with the Edinburgh Inventory (EHI) and that participants with a score below 30 were classified as left-handers (Methods, first section). As handedness was used as covariate in all analyses (and not just as descriptive information), I feel it would be beneficial to get a bit more information on the handedness classification applied by the authors. Firstly, given that no additional information is provided, I assume that the variable "Handedness" from the restricted HCP data was used. This variable is, however, problematic. That is, the HCP testing included only 9 of the 10 hand preference questions (leaving out "drawing") from the EHI but included the "footedness" question (see e.g. Ruck & Schoenemann, 2020, Laterality: DOI: [10.1080/1357650X.2020.1866001](https://doi.org/10.1080/1357650X.2020.1866001)) and it appears that the HCP variable Handedness is calculated by including "footedness" as the 10th item (at least that is what this preprint claims: doi: [10.1101/2022.11.14.516402](https://doi.org/10.1101/2022.11.14.516402)). Why this is the case, is not clear, but it likely is an "oversight" in HCP data handling. But as handedness and footedness are far from perfectly correlated (see doi: [10.1038/s41598-020-71478-w](https://doi.org/10.1038/s41598-020-71478-w)), "HCP Handedness" cannot be used to determine hand preference without leaving doubts. Thus, if the authors indeed have used the HCP handedness variable, this

requires (and I hate writing this) reanalysis of the data with a corrected handedness variable. Alternatively, if the authors were aware of the problem and retrieved the item-wise EHI answers from HCP and calculated an “own” laterality score, this information needs to be provided to the reader. In this case, it would also be interesting to hear how this was done, and how the laterality score was scaled. And why “30” was considered the correct cut-off for separating left and right handers. Considering the standard range from -100 to 100 suggested by the EHI, “30” appears somewhat arbitrary as a cut-off has it would actually include individuals with a slight right hand preference in the left hander sample.

We thank the reviewer for highlighting this important issue concerning the “Handedness” variable. In our analyses, we used the variable “Handedness” provided by the restricted HCP data (see Fig. 1A). We added a sentence in the *Method* (HCP participants) section to specify this point:

*“The values for Edinburg inventory scores can be found in the HCP database's restricted demographic file under the variable name “Handedness”, which is accessible only to authorized users. The reported results are consistent across alternate handedness metrics¹²³ (see supplementary material: *Replication of results using a different Edinburgh score* section).”*

HCP’s EHI score is based on (Schachter, Ransil & Geschwind, 1987, *Neuropsychologia*; DOI: [10.1016/0028-3932\(87\)90137-0](https://doi.org/10.1016/0028-3932(87)90137-0)), and includes the “footedness” item. Following reviewer recommendations, we computed an “unbiased” EHI score based on Raaf & Westerhausen's (2020, bioRxiv) preprint. This corrected EHI is computed the same way, excluding the “footedness” item, and doubling the weight of the “writing” item (see Fig. 1B). Both EHIs are highly and significantly correlated ($r=0.99$, $p<10^{-4}$, Fig. 1C), and there were no significant differences between them (Student's t-test: $\mu_{\text{difference}}=-1.21$, $t=-0.60$, $p=0.55$): highlighting the consistency of both metrics.

We reanalyze the data with the corrected EHI (Raaf & Westerhausen, 2022, bioRxiv) as a continuous covariate to respond to the reviewer’s concerns. The first re-analysis concerning the five features used for participants’ classification showed similar results (see Table 1) to the one using the Handedness variable provided by the HCP.

Second, the new analysis of the main effect of language lateralization on gradient asymmetry (see Table 2) showed a broadly consistent pattern with minor differences concerning the networks with a significant effect after the Bonferroni

correction. The asymmetry of the first gradient of the dorsal attentional network was no longer significant, and the same is true for the asymmetries of the second gradient of the default and salience/ventral attention networks. Critically, these results are concordant with our original analysis, highlighting the impact of language lateralization on the macroscale functional organization of the associative cortex. Finally, the new analyses using the corrected EHI as a continuous variable confirmed that the lateralization of the language network ($h^2=12.2\%$, $SE=7\%$, $p=0.028$), and the hemispheric asymmetries in gradients organization are heritable (see Table 3). Furthermore, heritability at the network level remained consistent: heritability was more significant within heteromodal association cortices than within unimodal networks. We kept the original handedness variable in the analysis since both EHIs are highly correlated, and the results are consistent across approaches. Again, we thank the reviewer for highlighting this issue and allowing us to make our results more robust with these new analyses.

Concerning the handedness threshold, as highlighted in the study provided by the reviewer (Ruck & Schoenemann, 2020, Laterality.), as well as in the referenced study in the *Method* section (Hervé et al., 2006, NeuroImage; DOI: [10.1016/j.neuroimage.2005.08.031](https://doi.org/10.1016/j.neuroimage.2005.08.031)), there exists a discrepancy between self-reported handedness and dimensional measures of handedness, such as EHI. Some right-handers will have a negative EHI score, while some left-handers will have a positive score. As proposed in Hervé et al. (2006, NeuroImage; DOI: [10.1016/j.neuroimage.2005.08.031](https://doi.org/10.1016/j.neuroimage.2005.08.031)), Mazoyer et al. (2014, PloS one; DOI: [10.1371/journal.pone.0101165](https://doi.org/10.1371/journal.pone.0101165)), and Robinson (2021, Encyclopedia of Autism Spectrum Disorders; DOI: [10.1007/978-1-4419-1698-3_877](https://doi.org/10.1007/978-1-4419-1698-3_877)), a cut-off of 30 at least in part, accounts for discrepancies between individuals' manual preference self-reports and their EHI score.

Figure 1. Hand preference assessment in the HCP. **(A)** Edinburgh Inventory (EHI) scores distribution as provided by the HCP. **(B)** Edinburgh Inventory score distribution proposed by Raaf & Westerhausen (2020, bioRxiv). **(C)** Correlation between provided EHI score and proposed corrected EHI score. The EHI scores correlate significantly ($r=0.99$, $p<10^{-4}$). Green dots are right-hander (RH) participants as defined in the paper, red ones are left-hander (LH). The blue line represents the linear regression line, and the blue light interval around the regression line represents the 95% confidence interval of the regression line.

Variable	Language lateralization phenotype (LLP) effect (mean \pm CI _{95%})	Post-hoc (p-value)	p-value
β_{Network} asymmetry	$\mu_{\text{strong typical}} = 1.79 \pm 0.07$	$\mu_{\text{strong typical}} > \mu_{\text{mild typical}}: <1.10^{-4}$	$p_{\text{model}}: <1.10^{-4}$
	$\mu_{\text{mild typical}} = 0.78 \pm 0.08$	$\mu_{\text{mild typical}} > \mu_{\text{atypical}}: <1.10^{-4}$	$p_{\text{LLP}}: <1.10^{-4}$
	$\mu_{\text{atypical}} = -0.74 \pm 0.19$	$\mu_{\text{strong typical}} > \mu_{\text{atypical}}: <1.10^{-4}$	
β_{Hubs} asymmetry	$\mu_{\text{strong typical}} = 2.79 \pm 0.10$	$\mu_{\text{strong typical}} > \mu_{\text{mild typical}}: <1.10^{-4}$	$p_{\text{model}}: <1.10^{-4}$
	$\mu_{\text{mild typical}} = 1.27 \pm 0.11$	$\mu_{\text{mild typical}} > \mu_{\text{atypical}}: <1.10^{-4}$	$p_{\text{LLP}}: <1.10^{-4}$
	$\mu_{\text{atypical}} = -0.89 \pm 0.27$	$\mu_{\text{strong typical}} > \mu_{\text{atypical}}: <1.10^{-4}$	
Inter-hemispheric r_z	$\mu_{\text{strong typical}} = 0.60 \pm 0.01$	$\mu_{\text{strong typical}} > \mu_{\text{mild typical}}: <1.10^{-4}$	$p_{\text{model}}: <1.10^{-4}$
	$\mu_{\text{mild typical}} = 0.49 \pm 0.01$	$\mu_{\text{mild typical}} < \mu_{\text{atypical}}: <1.10^{-4}$	$p_{\text{LLP}}: <1.10^{-4}$
	$\mu_{\text{atypical}} = 0.63 \pm 0.03$	$\mu_{\text{strong typical}} = \mu_{\text{atypical}}: 0.084$	
Strength asymmetry	$\mu_{\text{strong typical}} = 1.00 \pm 0.07$	$\mu_{\text{strong typical}} > \mu_{\text{mild typical}}: <4.10^{-3}$	$p_{\text{model}}: <1.10^{-4}$
	$\mu_{\text{mild typical}} = 0.83 \pm 0.07$	$\mu_{\text{mild typical}} > \mu_{\text{atypical}}: <1.10^{-4}$	$p_{\text{LLP}}: <1.10^{-4}$
	$\mu_{\text{atypical}} = 0.14 \pm 0.14$	$\mu_{\text{strong typical}} > \mu_{\text{atypical}}: <1.10^{-4}$	
Strength sum	$\mu_{\text{strong typical}} = 12.17 \pm 0.21$	$\mu_{\text{strong typical}} > \mu_{\text{mild typical}}: <1.10^{-4}$	$p_{\text{model}}: <1.10^{-4}$
	$\mu_{\text{mild typical}} = 9.37 \pm 0.22$	$\mu_{\text{mild typical}} < \mu_{\text{atypical}}: <1.10^{-4}$	$p_{\text{LLP}}: <1.10^{-4}$
	$\mu_{\text{atypical}} = 12.37 \pm 0.54$	$\mu_{\text{strong typical}} = \mu_{\text{atypical}}: 0.77$	

Table 1. ANCOVA results of the 5 functional features to classify participants. Significance of the Language Lateralization Phenotype main effect.

Network	Language lateralization phenotype (LLP) effect (mean \pm CI _{95%})	Post-hoc: t (p -value)	p -value	
G1	Default *	$\mu_{\text{typical}} = 1.99 \pm 0.28$	$\mu_{\text{typical}} > \text{atypical}: -4.24 (<.0001)$	$p_{\text{model}}: <.0001$ $p_{\text{LLP}}: <.0001$
	Control *	$\mu_{\text{typical}} = -0.30 \pm 0.30$ $\mu_{\text{typical}} = -5.78 \pm 0.34$	$\mu_{\text{typical}} > \text{atypical}: -4.56 (<.0001)$	$p_{\text{model}}: <.0001$ $p_{\text{LLP}}: <.0001$
	Limbic	$\mu_{\text{typical}} = 3.31 \pm 0.43$ $\mu_{\text{typical}} = 1.19 \pm 1.19$	$\mu_{\text{typical}} > \text{atypical}: -2.51 (0.1187)$	$p_{\text{model}}: 0.0424$ $p_{\text{LLP}}: 0.0121$
	Sal/VentAttn *	$\mu_{\text{typical}} = 1.20 \pm 0.29$	$\mu_{\text{typical}} > \text{atypical}: -4.35 (0.2205)$	$p_{\text{model}}: <.0001$ $p_{\text{LLP}}: <.0001$
	DorsAttn	$\mu_{\text{typical}} = -1.28 \pm 1.08$ $\mu_{\text{typical}} = 4.17 \pm 0.31$ $\mu_{\text{typical}} = 2.71 \pm 1.14$	$\mu_{\text{typical}} > \text{atypical}: -2.42 (0.0121)$	$p_{\text{model}}: 0.0004$ $p_{\text{LLP}}: 0.0157$
	Som/Motor	$\mu_{\text{typical}} = 0.03 \pm 0.03$ $\mu_{\text{typical}} = -0.05 \pm 0.05$	$\mu_{\text{typical}} > \text{atypical}: -0.31 (0.0717)$	$p_{\text{model}}: 0.7619$ $p_{\text{LLP}}: 0.7562$
	Visual	$\mu_{\text{typical}} = 1.74 \pm 0.15$ $\mu_{\text{typical}} = 1.17 \pm 0.55$	$\mu_{\text{typical}} > \text{atypical}: -1.97 (0.1847)$	$p_{\text{model}}: 0.0010$ $p_{\text{LLP}}: 0.0491$
	G2	Default	$\mu_{\text{typical}} = -0.51 \pm 0.15$ $\mu_{\text{typical}} = -0.95 \pm 0.55$	$\mu_{\text{typical}} > \text{atypical}: -1.52 (0.7085)$
Control *		$\mu_{\text{typical}} = -2.76 \pm 0.21$ $\mu_{\text{typical}} = -3.97 \pm 0.76$	$\mu_{\text{typical}} > \text{atypical}: -3.01 (0.0157)$	$p_{\text{model}}: 0.0002$ $p_{\text{LLP}}: 0.0026$
Limbic		$\mu_{\text{typical}} = -0.47 \pm 0.28$ $\mu_{\text{typical}} = -1.44 \pm 1.02$	$\mu_{\text{typical}} > \text{atypical}: -1.80 (0.7562)$	$p_{\text{model}}: 0.4801$ $p_{\text{LLP}}: 0.0717$
Sal/VentAttn		$\mu_{\text{typical}} = 1.39 \pm 0.26$ $\mu_{\text{typical}} = 0.67 \pm 0.67$	$\mu_{\text{typical}} > \text{atypical}: -1.44 (0.0491)$	$p_{\text{model}}: 0.0292$ $p_{\text{LLP}}: 0.1500$
DorsAttn		$\mu_{\text{typical}} = 1.27 \pm 0.31$ $\mu_{\text{typical}} = 1.37 \pm 1.13$	$\mu_{\text{typical}} < \text{atypical}: 0.16 (0.5987)$	$p_{\text{model}}: 0.1096$ $p_{\text{LLP}}: 0.8737$
Som/Motor		$\mu_{\text{typical}} = -2.30 \pm 0.24$ $\mu_{\text{typical}} = -1.15 \pm 0.87$	$\mu_{\text{typical}} < \text{atypical}: 2.50 (<.0001)$	$p_{\text{model}}: 0.0291$ $p_{\text{LLP}}: 0.0124$
Visual		$\mu_{\text{typical}} = -0.14 \pm 0.14$ $\mu_{\text{typical}} = -0.11 \pm 0.11$	$\mu_{\text{typical}} < \text{atypical}: 0.10 (0.0124)$	$p_{\text{model}}: 0.1747$ $p_{\text{LLP}}: 0.9213$
G3		Default *	$\mu_{\text{typical}} = -2.02 \pm 0.42$	$\mu_{\text{typical}} < \text{atypical}: 10.27 (0.9344)$
	Control *	$\mu_{\text{typical}} = 6.44 \pm 1.56$ $\mu_{\text{typical}} = -1.72 \pm 0.52$	$\mu_{\text{typical}} < \text{atypical}: 8.63 (0.1487)$	$p_{\text{model}}: <.0001$ $p_{\text{LLP}}: <.0001$
	Limbic *	$\mu_{\text{typical}} = 7.01 \pm 1.92$ $\mu_{\text{typical}} = -2.83 \pm 0.41$	$\mu_{\text{typical}} < \text{atypical}: 4.47 (0.0288)$	$p_{\text{model}}: <.0001$ $p_{\text{LLP}}: <.0001$
	Sal/VentAttn *	$\mu_{\text{typical}} = 0.74 \pm 0.74$ $\mu_{\text{typical}} = -5.00 \pm 0.37$	$\mu_{\text{typical}} < \text{atypical}: 4.71 (0.3072)$	$p_{\text{model}}: <.0001$ $p_{\text{LLP}}: <.0001$
	DorsAttn	$\mu_{\text{typical}} = -1.65 \pm 1.35$ $\mu_{\text{typical}} = -1.08 \pm 0.39$ $\mu_{\text{typical}} = 0.85 \pm 0.85$	$\mu_{\text{typical}} < \text{atypical}: 2.55 (0.0409)$	$p_{\text{model}}: 0.0116$ $p_{\text{LLP}}: 0.0108$
	Som/Motor	$\mu_{\text{typical}} = 0.29 \pm 0.20$ $\mu_{\text{typical}} = 1.14 \pm 0.73$	$\mu_{\text{typical}} < \text{atypical}: 2.19 (<.0001)$	$p_{\text{model}}: 0.0053$ $p_{\text{LLP}}: 0.0288$
	Visual	$\mu_{\text{typical}} = 0.37 \pm 0.16$ $\mu_{\text{typical}} = 0.82 \pm 0.60$	$\mu_{\text{typical}} < \text{atypical}: 1.45 (0.0171)$	$p_{\text{model}}: 0.0165$ $p_{\text{LLP}}: 0.1487$

Table 2. ANCOVA's results of the three gradient asymmetries values. Significance of the Language Lateralization Phenotype main effect. Colored lines display significant uncorrected effects ($p \leq 0.05$). Bold lines with a star are significant after Bonferroni correction for network number ($p \leq 0.007$).

	Network	Heritability (h^2)	Standard Error	p -value
G1	Default *	0.21	0.06	0.0001
	Control *	0.19	0.06	0.0014
	Limbic	0.09	0.07	0.0884
	Sal/VentAttn *	0.25	0.06	<.0001
	DorsAttn *	0.19	0.06	0.0008
	Som/Motor	0.08	0.05	0.0537
	Visual	0.08	0.06	0.0810
G2	Default *	0.16	0.06	0.0034
	Control	0.13	0.06	0.0092
	Limbic	0.03	0.05	0.2784
	Sal/VentAttn	0.07	0.06	0.1139
	DorsAttn	0.02	0.06	0.3392
	Som/Motor	0.06	0.06	0.1692
	Visual	0.02	0.05	0.3178
G3	Default *	0.29	0.06	<.0001
	Control *	0.28	0.06	<.0001
	Limbic *	0.16	0.06	0.0023
	Sal/VentAttn *	0.22	0.06	<.0001
	DorsAttn *	0.19	0.06	0.0008
	Som/Motor	0.00	0.00	0.5000
	Visual	0.10	0.06	0.0501

Table 3. Heritability of individualized gradient values. Colored lines reflect networks with significant heritability of gradient laterality ($p \leq 0.05$ uncorrected). Lines with a star remain significant after Bonferroni correction ($p \leq 0.007$).

Secondly, if I understand the methods description correctly, 3 of the 5 features that are used for the asymmetry phenotype classification are derived from the HCP rs-fMRI, while the same rs-fMRI was utilized to derive the 3 gradient asymmetries. That means that the rs-fMRI data contributed (although with different processing steps and to different degree) both to determining the independent and dependent variable used in the second set of analyses (presented under “Gradients asymmetries and atypical lateralization”). Consequently, it is not clear whether the found “gradient” differences between the phenotype groups, simply reflect that the groups were determined (partly) by the same data. Thus, the authors need to convince that this is not the case.

The reviewer is correct, 3 of the five features (*Inter-hemispheric r_z* , *Strength asymmetry*, and *Strength sum*) to classify participants, and the 3 gradients are derived from the same rs-fMRI data. To be precise and as described in the *Method* section, the 3 features used for the classification are metrics that have been shown to characterize the language network (Labache et al., 2019, *Brain Struct Funct*; DOI: [10.1007/s00429-018-1810-2](https://doi.org/10.1007/s00429-018-1810-2), Labache et al., 2020, *eLife*; DOI: [10.7554/eLife.58722](https://doi.org/10.7554/eLife.58722)). Of note, these metrics have been computed on a small subset of the full brain connectivity matrices at rest, *i.e.*, the 18x18 matrices

describing the correlations at rest between each of the 18 brain regions composing the language network in the left, and right hemispheres. Two of these metrics (*Strength asymmetry*, and *Strength sum*), therefore, use only 0.42% ($=\frac{18 \times 17 \times 2}{384 \times 383}$) of all the available information to be computed. This proportion is even weaker for the *Inter-hemispheric r_z* since this metric is only computed on the 18 inter-hemispheric homotopic correlations between the language regions, *i.e.*, 0.01% ($=\frac{18}{384 \times 383}$) of the available data. On the other hand, the computation of the 3 gradients is based on the whole brain connectivity matrices, *i.e.*, the 384x384 correlation matrices.

Furthermore, the 3 resting-state features used in the classification are essential to characterize atypical language organization. They have highlighted the specificity of language network organization at rest, with atypical individuals having an original symmetrical language organization. At the same time, typical remains leftward asymmetrical at rest (Labache et al., 2020, eLife; DOI: [10.7554/eLife.58722](https://doi.org/10.7554/eLife.58722)). On the other hand, the gradients provide a different kind of information: the functional topography of the cortex, and its sensory integration flow, allowing us to know which parts of the brain are heteromodal associative, unimodal, primary, etc.

To conclude, the found gradient differences between the phenotypes are unlikely because of the small data overlap between the computation of each metric.

*Finally, I feel the manuscript is densely written and could benefit from thorough language editing. Many sentences are (at least in my opinion) unnecessarily complicated and make it difficult to understand what the authors mean. One of several examples: “**The parallel and interdigitated organization of cortical networks suggests that the language system may impinge upon, and be influenced by, putatively distinct yet spatially adjacent networks**” (top of 2nd page of discussion). As nice as this sounds, does this not just say that language processing relies on cortical networks that may interact (which is not surprising).*

We apologize for the lack of clarity in our initial submission. Here we were hoping to communicate that, due to the complex nature of brain organization, systems not explicitly related to the language network may be tied to language network lateralization. We have edited the revised manuscript, working to reduce the density of our writing.

The sentence highlighted by the reviewer now reads as follows:

“Cortical networks exhibit a parallel and tightly interdigitated organizational structure. As a result, the language system may impinge upon, and be influenced by, putatively distinct yet spatially adjacent networks.”

We also rewrite the following sentences:

- *Introduction* section; “The hemispheric specialization of a range of specific functions [...] and posterior cortices”, now reads: “The hemispheric specialization of a range of specific functions has been well characterized. One of the most widely investigated is the left-lateralized high-order language network encompassing aspects of the anterior and posterior cortices.”

REVIEWER 2

The study uses data from the Human Connectome Project (HCP) to quantify hemispheric language dominance based on task fMRI, and relate it to hemispheric differences in macroscale functional gradients based on resting fMRI. In addition, the study shows heritability of language laterality and gradient asymmetries using HCP twin data. The findings contribute to understanding how specific task-functional organization can relate to broader aspects of brain network organization. The findings might also provide a route to using large-scale resting fMRI data to perform molecular genetic studies of hemispheric language dominance. I find the study clearly described and a useful contribution to the field. Further consideration could be given to the following issues:

- The variation across individuals is quantitative and continuous, but the authors rely on clustering to create groups (categories) for subsequent analyses. The particular clusters that were defined do not appear especially robust compared to alternative cluster solutions. The issue could be avoided by treating the data as continuous. If not, then a stronger rationale could be given for the cluster-based approach, together with clearer indications that the chosen solution was robust compared to others.

We agree with the referee's comment.

First, to test the robustness of the chosen cluster solution, we conducted a new hierarchical cluster analysis with multiscale bootstrap resampling (number of bootstraps 10,000, using Euclidean distance as metrics and Ward's criterion as linkage criteria. Shimodaira, 2002, Systematic Biology; DOI: [10.1080/10635150290069913](https://doi.org/10.1080/10635150290069913)) using the *pvclust* R library (Suzuki et al., 2022,

pvclust: R package version 2.2.0). This package provides, for each dendrogram partition, a p -value called Approximately Unbiased (AU) p -value. This p -value represents the probability of a given partition occurring among the HCP sample, and indicates the partition's reliability.

The Approximately Unbiased p -value for 3 clusters is 0.98 ($CI_{95\%} = \pm 0.002$), leading us to reject the hypothesis that the clusters are unstable.

We also emphasize that we did not *a priori* decide the number of clusters. Instead, the optimal number of clusters for this multivariate classification was obtained using a fully unsupervised methodology and a combination of 26 statistical criteria. These indices are available through the R package *NbClust* (Charrad et al., 20, Journal of Statistical Software; DOI: 10.18637/jss.v061.i06, R package version 3.0.1) and evaluate the stability of different clustering schemes. The final number of clusters is selected according to the majority rule. For the present data, the first result is 4 clusters with 11 indices, followed by 3 clusters with 5 indices (see Table 4). Choosing 4 clusters would have divided the Strong Typical group into two. This partition was not desirable in this specific study. Indeed, the point of the paper was not to over-characterize every variation of language lateralization but rather to study the relationship between typical or atypical language lateralization and its link with a potential global shift in cortical organization. Subdividing the Strong Typical group would not have added anything interesting to the present study. We then select the second-best partition: 3 clusters (see Table 4). Of note, this clustering approach replicates our previous results (Labache et al., 2020, eLife) using a completely different database, highlighting the strengths of the present multi-modals classification and phenotype description.

It has been shown in the literature (Mazoyer et al., 2014, PloS one; DOI: 10.1371/journal.pone.0101165) that the distribution of lateralization for sentence production, although of continuous nature, could be used to classify individuals into 3 discrete categories. It is also true for lateralization indexes derived directly from the resting-state data (Joliot et al., 2016, Neuropsychologia; DOI: 10.1016/j.neuropsychologia.2016.03.013). Furthermore, our previous results (Labache et al., 2020, eLife; DOI: 10.7554/eLife.58722) have shown that these 3 cluster solution was relevant and allowed us to reveal the non-trivial organization of atypical individuals that does not mirror that of typical individuals. That an unsupervised classifier succeeded at characterizing this multivariate dataset in 3 discrete categories only constitutes a validation of our approach. As pointed out by the reviewers, classification methods based on multiple

continuous variables almost inevitably lead to clusters that overlap at their boundaries for some of these variables (Bishop, 2006, Pattern recognition and machine learning; ISBN: 978-1-4939-3843-8). Such overlaps do not invalidate the classification approach. Rather, they reflect the importance of simultaneously accounting for all variables instead of looking at each separately.

We finally added/modified the following explanation and justification for the classification at the end of the *Language Lateralization Identification* section (see *Method - Section Analyses* section):

“We employed an unsupervised methodology to determine the optimal number of clusters. Combining results from the R package *NbClust* (R package version: 3.0.1) and *pvclust* (R package version: 2.2.0), we selected a 3-cluster solution (strong typical, mild typical, and atypical), which was shown to reproduce our previous results. *pvclust* showed that the 3-cluster solution was stable (Approximately Unbiased p -value= 0.98, $CI_{95\%} = \pm 0.002$). Furthermore, among the 26 indices used by *NbClust* to evaluate the stability of different clustering schemes, the 3-cluster partition was the second most supported, with 5 indices in its favor. The most supported partition was a 4-cluster solution with 11 indices. This would have led to the partitioning of the strong typical group, which was deemed unnecessary as it would not have provided new insights to the aim of this study.”

# of clusters	2	3	4	6	8	10	13	15
# of indices supporting the cut	1	5	11	2	2	1	2	2

Table 4. Determination of clusters' number. For each of the 26 indices proposed by the R package *NbClust* (Charrad et al., 20, Journal of Statistical Software; DOI: 10.18637/jss.v061.i06, R package version 3.0.1), the best number of clusters is reported in the table. The best clustering schemes from the different results were obtained by varying all combinations of the number of clusters, distance measures, and clustering methods.

- Hemispheric language dominance is most pronounced for language production tasks. The particular task contrast used in HCP involves comparing comprehension of brief narratives with an arithmetic task. The Discussion would benefit from some consideration of this issue. Was the HCP task optimal for determining hemispheric language dominance?

We thank the reviewer for their pertinent comment and would like to briefly justify our use of the listening task in this study.

The reviewer is right about the most pronounced language dominance during a production task. However, we have previously shown that even if a listening task is less leftward lateralized than a production (the most lateralized) and a reading task, it is still significantly activated and leftward asymmetrical (Labache et al., 2019, *Brain Struct Funct*; DOI: [10.1007/s00429-018-1810-2](https://doi.org/10.1007/s00429-018-1810-2)). Similarly, Bruckert et al. (2021, *Laterality*; DOI: [10.1080/1357650X.2021.1898416](https://doi.org/10.1080/1357650X.2021.1898416)) have shown that the lateralization strength of different language tasks (semantic association and word generation) is significantly correlated, supporting the view that language lateralization is multifactorial. Importantly, our previous study (Labache et al., 2020, *eLife*; DOI: [10.7554/eLife.58722](https://doi.org/10.7554/eLife.58722)) has demonstrated that a comprehensive approach to identifying language lateralization phenotype, which incorporates both task-based fMRI and resting-state fMRI metrics, is essential in detecting atypical individuals who may not be identified through traditional task-based fMRI alone.

Furthermore, the classification includes asymmetries during the language task at the network level (i.e., the full language network) and the hub level (corresponding to Broca's and Wernicke's areas). We showed in the same study (Labache et al., 2020, *eLife*; DOI: [10.7554/eLife.58722](https://doi.org/10.7554/eLife.58722)) that the asymmetries at the hub level are amplified and therefore optimize the detection of language dominance. It has also been shown that asymmetries measured during language tasks compared to a high reference condition are an appropriate marker of language dominance (Binder et al., 2011, *Radiology*; DOI: [10.1148/radiol.11101344](https://doi.org/10.1148/radiol.11101344)).

We believe the HCP task, combined with resting-state lateralization features, is optimal for determining hemispheric language dominance.

Accordingly, we added the following paragraph to the *Discussion* section:

“From a methodological perspective, the HCP database contains only one language comprehension task (Barch et al., 2013). Of note, brain responses to language comprehension have been shown to share consistent lateralization patterns with other language tasks (Bruckert et al., 2021). Furthermore, incorporating asymmetries at the hubs (Broca's and Wernicke's areas) increases the amplitude of asymmetries in the comprehension task. Finally, our prior work

showed that combining resting and task metrics is essential to accurately identify atypical individuals(Labache et al., 2020).”

- *The study reports heritability of lateralization of the language network, and also heritability of hemispheric asymmetries in gradient organization. The same data and software could be used to assess the genetic correlation between these two types of measure, i.e. to what extent do the same genetic variants affect hemispheric language dominance and asymmetries of gradient organization, versus being independently heritable (and therefore correlated for non-genetic reasons).*

We thank the reviewer for highlighting this approach. This is an area of focused interest in our group, and we also considered this possibility. However, with our limited sample size, performing accurate genetic correlation analyses is not feasible. Here, we hope to explore the shared genetic bases of cortical lateralization in our subsequent work.

- *For introducing/discussing the genetic parts it would be relevant to cite recent genetic association studies of left-handedness and brain asymmetry:*

1. <https://www.nature.com/articles/s41562-020-00956-y>
2. <https://www.nature.com/articles/s41562-021-01069-w>
3. <https://www.pnas.org/doi/10.1073/pnas.2113095118>
4. <https://www.nature.com/articles/s41598-019-42515-0>
5. <https://academic.oup.com/brain/article/142/10/2938/5556832>

These citations have been added to the revised manuscript.

References 1, 4, and 5 have been added in the revised manuscript:

“Those results concordant with recent development highlighting that handedness results from a complex combination of polygenetic traits (Cuellar-Partida et al. 2021; de Kovel and Francks 2019): each increasing the odds of being left-handed. Knowing moreover that atypical individuals are mainly left-handers, a complex picture is then drawn, where a multiplex association of biological pathways may result in the atypical phenotype (Wiberg et al. 2019; Carrion-Castillo et al. 2019).”

References 2, and 3 have been added to the *Results* section (*Heritability of language lateralization and gradient asymmetry*):

“Other studies have highlighted the association between differences in structural asymmetry in left- and right-handed people and the genetic basis of their manual

preference(Sha et al. 2021, PNAS), as well as the genetic basis of structural asymmetries(Sha et al. 2021, Nat. Hum. Behav.).”

REVIEWER 3

Leveraging a higher-order language atlas and following previous work, the authors performed a hierarchical clustering based on a combination of resting-state and task-evoked fMRI features. They identified 3 subgroups of subjects related to language lateralization in the HCP database. Gradients were computed through diffusion map embedding on functional connectivities among the AICHA atlas ROIs. Asymmetry of each gradient in typical subjects was calculated as the difference between gradient values in the left and right hemispheres for each canonical network.

Then, the impact of language lateralization on the asymmetry of each canonical network was assessed using an ANCOVA, comparing atypical subjects for language lateralization and typical and mild typical subjects merged into a single group. Finally, using a multidimensional heritability analysis, genetic contribution to the phenotypic variance of language network lateralization, on the one hand, and the hemispheric asymmetries in gradient organization, on the other hand, was assessed. In addition, the heritability of gradient asymmetry values for each canonical network was estimated.

The manuscript is well-written, and the subject is highly interesting and original. Many analyses have been carried out, using various methods and uncovering valuable results. However, I would have some questions about several points.

The experimental design of this work is very complex, and for the sake of clarity, the author could provide a diagram of the relationships between the different analyses and results.

We thank the reviewer for their suggestion. We have added an experimental design figure to the *Supplementary Materials - Experimental workflow* section. Please see Figure 2 below (Supplementary Fig. 2 in the manuscript).

We added the following sentence in the *Method - Statistical Analyses* section to refer to the figure:

“An overview of our experimental workflow is shown in Supplementary Fig. 2.”

Figure 2. Experimental workflow. **(A)** Different parcellations used. Box 1 (top to bottom): surface mask of the core high-order language atlas (SENSAAS, see Method – Language Atlas Statistics section for a full description), hubs of SENSEAAS, SENSEAAS in volume space. Box 2 (top to bottom): AICHA atlas, 7 canonical resting-state networks (see Method – Connectivity Embedding section for a full description). **(B)** Brain metrics used to characterize the participants (HCP, $n=995$). The 5 functional metrics of box 1

are fully described in the Method – Language Atlas Statistics section. The gradient embedding metrics presented in box 2 are fully described in the Method – Connectivity Embedding section. (C) Statistical analysis and their goals. Box 1: identification of the different language phenotypes is fully described in the Method – Statistical Analyses: Language Lateralization Identification section. Box 2: the analysis of covariance to analyze the impact of language lateralization on the gradient embedding is fully described in the Method – Statistical Analyses: Language Lateralization impact on gradient asymmetry section, the heritability analysis is fully described in the Method – Statistical Analyses: Heritability of Gradient Asymmetry and Language Lateralization Phenotype section. Only the left hemisphere of each parcellation and metrics is shown for illustration purposes.

When identifying subgroups of subjects based on their language network lateralization using hierarchical clustering, the number of subgroups was set to 3, based on previous work (Labache et al. Elife 2020). In this previous work, the cohort was enriched for left-handed subjects with no twin pairs or siblings, which is not the case in HCP. Moreover, input features based on task-fMRI data were derived from production, reading and listening tasks versus a Story-Math contrast in the current study. I wonder if the 3-subgroup result in Labache et al. Elife 2020 could be directly applied here, without any machine learning optimization of this hyperparameter, considering that the input is different? It would be valuable to reproduce this result in similar but not identical conditions.

We thank the reviewer for their comment on how the 3 groups were identified, a point also made by the previous reviewer. The 3-subgroup result from Labache et al. (2020, eLife) can be directly applied here for the following reasons.

One of our working hypotheses was that one language-related task combined with the resting-state laterality markers would be enough to identify atypical individuals, specifically since we have shown that the language tasks' lateralization was correlated with each other (see previous reviewer's answer). The present results show that one t-fMRI lateralization marker combined with the resting-state ones is sufficient to identify atypical individuals. Indeed, the phenotype characterization of each group is similar to the ones identified in Labache et al. (2020, eLife; DOI: [10.7554/eLife.58722](https://doi.org/10.7554/eLife.58722)).

There is no hyperparameter to optimize in the unsupervised classification method we used (hierarchical agglomerative classification). The hierarchical classification methods compute the similarity between the characteristics of the individuals and lead to the establishment of a dendrogram illustrating the relative position of an

individual compared to the others according to their similarities. The only "parameter" to choose is the number of groups or clusters. Considering the current commentary, and the previous one, we first analyzed the stability of our partition (3-cluster partition) using a bootstrapping classification method (Shimodaira, 2002, Systematic Biology; DOI: [10.1080/10635150290069913](https://doi.org/10.1080/10635150290069913)). This analysis confirmed the stability of the dendrogram and the 3-cluster partition solution. A second assessment of the stability of the dendrogram based on a total of 26 statistical indices (Charrad et al., 20, Journal of Statistical Software; DOI: [10.18637/jss.v061.i06](https://doi.org/10.18637/jss.v061.i06)) also led to the selection of 3 clusters. Ultimately, the number of groups is based on the literature (Mazoyer et al., 2014, PloS one; DOI: [10.1371/journal.pone.0101165](https://doi.org/10.1371/journal.pone.0101165), and Labache et al., 2020, eLife; DOI: [10.7554/eLife.58722](https://doi.org/10.7554/eLife.58722)) and a data-driven approach.

Considering the reviewer's remark about the HCP cohort having different characteristics than our previous work (Labache et al., 2020, eLife; DOI: [10.7554/eLife.58722](https://doi.org/10.7554/eLife.58722)), we choose to re-run a classification by only selecting unrelated individuals (HCP singleton, n=110, 92 right-handers) and then by making the sub-sample equally distributed for handedness (random selection of 18 right-handers individuals among the 92), resulting in a small sub-sample of 36 participants (18 left-handers). The classification results are presented in Figure 3. The 3-cluster partition solution shows similar results to the one using the full sample (n=995) and the one in Labache et al. (2020, eLife). Furthermore, participants are classified into the same groups as before.

Figure 3. Identification and characterization of language lateralization in a subsample of 36 HCP participants (18 left-handers) with the same demographic characteristics as in Labache et al. (2020, eLife; DOI: [10.7554/eLife.58722](https://doi.org/10.7554/eLife.58722)): unrelated individuals (singleton participants) and 50% of left-handers. **(A)** Hierarchical clustering resulted in the identification of three populations with varying degrees of language organization. The 3-cluster partition solution is composed as follows: strong typical individuals (n=16, in orange), mild typical (n=11, in purple), and atypical (n=9, in blue). **(B)** Resting-state metrics characterizing the language network: the average homotopic inter-hemispheric intrinsic correlation (r_z), and both the asymmetry ($\text{Strength}_{\text{asymmetry}}$) and the sum ($\text{Strength}_{\text{sum}}$) of the average strength. **(C)** Language task-related metrics characterizing the language network: the average BOLD asymmetries values in the story-math contrast at the network level ($\beta_{\text{Network asymmetry}}$) and hubs level ($\beta_{\text{Hubs asymmetry}}$). The error bars represent the 95% confidence interval.

Multiple test correction is not mentioned in the ANCOVA of language lateralization and the 5 input features of hierarchical clustering. Are the reported p-values corrected or not?

In supplementary tables 10 and 11, when correcting for multiple testing, the correction accounted for 7 networks, but there were also 3 gradients for each network; I wonder if the p-values should have been corrected for 21 tests instead of only 7, as in supplementary figure 2? Same question for ANCOVA of language lateralization and

lateralization of large-scale cortical organization.

Again, we thank the reviewer for their close reading of our manuscript, and we apologize for the lack of detail in our initial submission.

In the *Results - Identification of Atypically Lateralized Individuals for Language* section, the reported p -values are the corrected p -values for multiple comparisons using Tukey's range test. A p -value is then significant if less than 0.05. We added the following sentence in the *Methods - Language Lateralization Identification* section:

“Post-hoc analyses were conducted using Tukey's range test for multiple comparisons (to account for the number of language lateralization phenotypes: strong typical, mild typical, and atypical), or Student's t -test for binary ones. The reported p -values in the corresponding *Results* section are corrected for multiple comparisons.”

In the *Results - Gradient asymmetries and atypical lateralization* section, the reported p -values are not corrected for multiple comparisons. Then, a p -value is significant if less than $\frac{0.05}{7} = 0.007$. We added the following sentence in the *Methods - Language Lateralization impact on gradient asymmetry* section:

“The reported p -values in the corresponding *Results* section are not corrected for multiple comparisons and are significant if less than 0.007.”

We thank the reviewer for their essential comment about the multiple comparisons problem.

We wanted to highlight that we did not compare the gradients between each other in the *Results - Gradient asymmetries and atypical lateralization* section. In this case, we would therefore have had $n = \frac{21*(21-1)}{2} = 210$ comparisons to test. The Bonferonni correction for 21 tests was applied in (previous) Supplementary Figure 2 because we were comparing the 21 values of the correlation matrix to each other. Since the correlation matrix is symmetrical, and because we are using the seven canonical networks partition and have no information about the correlation of a network with itself, the number of tests to perform for comparison is equal to $n = \frac{7*(7-1)}{2} = 21$.

Thus, correcting the p -values for 21 comparisons is not appropriate in the cases mentioned above by the reviewer (*Results - Gradient asymmetries and atypical lateralization* section).

The authors conclude that genetic factors substantially influence the lateralization of both specific cognitive functions (language, I suppose) and the broad functional organization of the cortex. As covariance between language lateralization and canonical networks lateralization was assessed, the genetic correlation between these could also be computed to give even more insight into the genetic architecture of brain lateralization.

We appreciate the reviewer for drawing attention to this method. Our group is interested in this particular area and has considered this option. Nonetheless, due to the constraints of our sample size, conducting precise genetic correlation analyses is not viable at this time. Our future research aims to investigate the shared genetic foundations of cortical lateralization.

The supplementary figure 2 is never commented on.

We thank the reviewer for catching this error. Supplementary Figure 2 is no longer relevant to our argument. We have therefore removed it from the revised manuscript.

For clarity, the third paragraph of "Gradients asymmetries atypical lateralization" should refer to Supplementary table 10. Typo at the end of paragraph : dorsal-attentional ($\mu_{\text{typical}}=-0.61\dots$) \rightarrow dorsal-attentional ($\mu_{\text{L-R(typical)}}=-0.61\dots$)

We thank the reviewers for having detected this typo; we have corrected it.

We added this sentence in the abovementioned paragraph (Supplementary table 10 is now Supplementary table 7):

"See Supplementary Table 7 for a complete description of each network's typical gradient asymmetry values."

I would recommend this work for publication in Nature Communications with major revisions.

We thank the reviewer for their helpful comments. The document has been revised to incorporate all their comments.

ADDITIONAL AUTHOR CORRECTIONS

- To better reflect the content of the present article, we removed “Atypical” from the title. It is now read as “Language network lateralization is reflected throughout the macroscale functional organization of cortex”.
- We unified and detailed the nomenclature of mathematical symbols throughout the results section (for example, μ becomes μ_{hubs} when appropriate, etc.).
- We added some relevant references in the following:
 - *Introduction* section:
 - we added a reference to Gerrits 2022 (Neuropsychol Rev; DOI: [10.1007/s11065-022-09575-y](https://doi.org/10.1007/s11065-022-09575-y)), for this sentence “[...] the origins, mechanisms, and consequences of hemispheric specialization are still largely unknown^{11-14, Gerrits 2022}.”
 - the *Results* section (*Heritability of language lateralization and gradient asymmetry*):
 - “[...] the role of genetics in sculpting the lateralization of cognitive functions and associated asymmetries in the macroscale organization of cortex has remained largely unexplored. Prior work has established the heritability of gradient asymmetries in humans, an aspect of cortical organization likely present across non-human primates (Wan et al. 2022).”
 - the *Discussion* section:
 - “These results are consistent with mounting evidence suggesting an association between language lateralization and asymmetry in functional connectivity at rest (Joliot et al. 2016; Raemaekers et al. 2018), with local brain efficiency in atypical individuals (Wang et al. 2019), and with the presence of individual-specific functional deviations, or “network variants,” that systematically differ across hemispheres (Perez et al. 2023).”
- We added the following sentence to the *Method - Statistical Analyses* section:
 - “An overview of our experimental workflow is shown in Figure 5.”
- We added a new section to the *Supplementary Materials*: see *Replication of results using a different Edinburgh score*. This new section summarizes the results presented to reviewer 1 about the Edinburgh Inventory score.
- We harmonized through the text the denomination of the networks by using the expression “7 canonical networks”:

- in the legend of Figure 2: “[...] according to the 7 canonical networks identified in Yeo et al. [...]”
- in the legend of Figure 3: “[...] according to the 7 canonical networks [...]”
- in the *Methods* section (*Connectivity Embedding*): “[...] according to each of the 7 canonical networks described [...]”, and “[...] each AICHA region has been assigned to one of the 7 canonical networks [...]”
- We added precision to the sample size in the *Results* section (*Gradient asymmetries and atypical lateralization*), the legend of Figure 3, and Figure 4.

Reviewer #1 (Remarks to the Author):

The authors convincingly addressed most of my concerns and questions. The issue with the HCP handedness variable remains. That is, the authors nicely demonstrate the correlation of the HCP handedness variable with the corrected handedness score, as well as that the results of the analyses are more or less comparable. Having shown this, the authors keep on reporting the original analysis. However, I personally find this a bit inconsequential, as the used HCP variable handedness does (after all) NOT measure "handed preference" but a combination of "hand and foot preference", so that any following interpretation of the findings with regard to "handedness" is semantically wrong. Of course, the two indices correlate highly, as 9 out of the 10 items are identical. Importantly, however, the latent construct measured by the questionnaire is different. No statistical test can change this.

Beyond this fundamental problem, the authors do not show that the "corrected" handedness laterality score leads to the same classification (distribution) when forming hand preference groups.

Reviewer #2 (Remarks to the Author):

Many thanks to the authors for addressing my earlier points.

One point I raised was that hemispheric language dominance is most pronounced during language production tasks, but the HCP does not include such a task, which raises a question about whether HCP is optimal for studying hemispheric language dominance. In their rebuttal the authors make several valid points that support the use of HCP data for this purpose, but I think not all of these points made it into the revised manuscript. The authors can consider integrating the points more fully in the manuscript.

Regarding the genetics of handedness, the authors have added new text that includes 'handedness results from a complex combination of polygenetic traits'. As the majority of variation in handedness is not heritable, I suggest to modify the sentence to something like 'the genetic contribution to variation in handedness is complex and polygenic'.

Clyde Francks

Reviewer #3 (Remarks to the Author):

I thank the authors for their detailed response.

The authors have conveniently answered the reviewers' questions and modified the manuscript accordingly, which is now, in my opinion, suitable for publication in NComm.

RESPONSE TO REVIEWERS AND EDITORIAL OFFICE

We are deeply grateful for the acceptance, in principle, of our manuscript for publication in Nature Communications. We sincerely appreciate the invaluable feedback and constructive suggestions provided by the reviewers. Their comments have contributed greatly to refining our work, and address the remaining concerns and editorial requests in the attached documents. In addition to addressing the reviewers' remarks, we edited our manuscript to comply with Nature Communications' policies and formatting requirements. Once again, I thank the reviewers and the editorial team for their guidance and support throughout this process.

Detailed responses to each remark are included below. Editorial and reviewer comments are shown in *italicized font*, answers are shown in black, and new text is shown in blue.

REVIEWER 1

The authors convincingly addressed most of my concerns and questions. The issue with the HCP handedness variable remains. That is, the authors nicely demonstrate the correlation of the HCP handedness variable with the corrected handedness score, as well as that the results of the analyses are more or less comparable. Having shown this, the authors keep on reporting the original analysis. However, I personally find this a bit inconsequential, as the used HCP variable handedness does (after all) NOT measure "handed preference" but a combination of "hand and foot preference", so that any following interpretation of the findings with regard to "handedness" is semantically wrong. Of course, the two indices correlate highly, as 9 out of the 10 items are identical. Importantly, however, the latent construct measured by the questionnaire is different. No statistical test can change this.

We appreciate the reviewer's thorough evaluation of our paper. We acknowledge their concerns about the semantic differences between the original HCP handedness variable, which combines hand and foot preference, and the corrected handedness score focusing solely on hand preference. However, we have elected to retain the original analysis for two primary reasons: 1) We have demonstrated a strong correlation between the HCP handedness variable and the reviewer-suggested handedness score ($r=0.98$), and the results of the analyses are comparable. 2) Including the original analysis allows our work to maintain consistency with the existing literature that uses the HCP handedness variable.

We added the following sentence in the *HCP participants* sub-section (*Method* section):

“It is worth noting that the HCP's EHI score includes one foot-related item among its ten items. As a result, it does not solely measure manual preference. Analyses using a corrected EHI score (Raaf & Westerhausen, 2023) that excludes the foot-related item yield consistent results (see supplementary material: *Replication of results using a different Edinburgh score* section).”

Beyond this fundamental problem, the authors do not show that the "corrected" handedness laterality score leads to the same classification (distribution) when forming hand preference groups.

We appreciate the reviewer's concern regarding the "corrected" handedness laterality score and its potential impact on the classification. Please allow us to clarify our approach.

In our study, the handedness item does not play a role in the classification process. Instead, we rely on five functional language laterality metrics (network- and hubs-level asymmetry during the language task, sum and asymmetry of strength, and homotopic inter-hemispheric connectivity value at rest). This ensures that our classification is driven entirely by brain data, and as a result, any modifications to the handedness variable computation will not impact the classification outcomes.

We utilize the handedness metric as a demographic descriptor post-classification to verify that our classification results align with existing literature and to confirm that the atypical group contains a higher proportion of left-handers. To address the reviewer's concern, we have added a sentence in the *Replication of results using a different Edinburgh score* sub-section (found in the *Supplementary Materials* section) which describes the updated proportion of left- and right-handers using the corrected handedness metric:

“Using the same threshold as in the main analysis: an upper limit of 30 defines left-handedness. The unbiased EHI score leads to the following distribution: 110 left-handers and 885 right-handers. 3 individuals (2 mild typical and 1 strong typical) classified as right-handers with the HCP's EHI score are now left-handers; likewise, 3 previously right-handers (2 mild typical and 1 strong typical) are now left-handers using the unbiased EHI score. Finally, the distribution of

handedness among the groups does not change; 26 left-handers among 82 individuals in the atypical group, 48 left-handers among 433 in the mild typical group, and 36 left-handers among 480 individuals in the strong typical group.”

In the revised manuscript, we added the distribution of handedness in the main text (*Identification of Atypically Lateralized Individuals for Language* section):

“a strong typical group characterized by a strong leftward asymmetry during language task performance ($n=480$, 36 left-handers), a mild typical group with moderate leftward asymmetry ($n=433$, 48 left-handers), and atypical individuals showing a rightward asymmetry ($n=82$, 26 left-handers, Fig. 1B)”

REVIEWER 2

Many thanks to the authors for addressing my earlier points.

One point I raised was that hemispheric language dominance is most pronounced during language production tasks, but the HCP does not include such a task, which raises a question about whether HCP is optimal for studying hemispheric language dominance. In their rebuttal the authors make several valid points that support the use of HCP data for this purpose, but I think not all of these points made it into the revised manuscript. The authors can consider integrating the points more fully in the manuscript.

We are grateful for the reviewer's positive assessment of our manuscript, and we extend our thanks for their insights and guidance during the evaluation process.

In response to this comment, we have added the missing points to the discussion section alongside the previous points raised:

“Additionally, although an auditory task exhibits less lateralization than a production task (which is the most lateralized task) or a reading task, it still significantly activates and lateralizes the language network in the left hemisphere (Labache et al., 2019). In line with this, asymmetries observed during language tasks, compared to a high reference condition, serve as a suitable marker for determining language dominance (Binder et al., 2011).”

Regarding the genetics of handedness, the authors have added new text that includes ‘handedness results from a complex combination of polygenetic traits’. As the majority of variation in handedness is not heritable, I suggest to modify the sentence to

something like 'the genetic contribution to variation in handedness is complex and polygenic'.

We have updated the text in the revised manuscript in response to this comment.

REVIEWER 3

I thank the authors for their detailed response.

The authors have conveniently answered the reviewers' questions and modified the manuscript accordingly, which is now, in my opinion, suitable for publication in NComm.

We truly appreciate the reviewer's positive evaluation of our manuscript, and we thank their valuable insights and support throughout the review process.

ADDITIONAL AUTHOR CORRECTIONS

- Modifications induced by the Author Checklist:
 - we added an *Author list* section.
 - we added an *Author Contributions* section:

“L.L. and, A.J.H. designed the research. L.L. conducted the research. L.L., A.J.H, and T.G. analyzed and interpreted the results. L.L., and A.J.H. wrote the paper, which all authors commented on and edited. L.L. and A.J.H. made figures. L.L. analyzed the data. L.L. published the code. All authors provided analytic support.”

- we added the sample size in the legend of Figure 1c:

“Raincloud plots display the five functional metrics within each identified group ($n_{\text{strong typical}}=480$, $n_{\text{mild typical}}=433$, $n_{\text{atypical}}=82$).”

- we defined the measure of the center for the error bands of supplementary figure 3c.

“Vertical dashed line represents the left-handed definition threshold (<30, right-handed otherwise) for the HCP's EHI score. The horizontal dashed line represents the left-handed definition threshold (<30, right-handed otherwise) for the unbiased EHI score. The solid vertical and horizontal lines represent the handedness definition with a threshold of 0 (<0 defines left-handedness, right-hander otherwise)”

- we added the statistical test used in Supplementary Table 7, Figure 3d:

“Post-hoc analyses were conducted using a two-sided Student’s *t*-test.”

- we added the statistical test used in Supplementary Tables 2, 3, 4, 5, 6, 8, 9, 10, 12, 13:

“The displayed *p*-values correspond to that of the *F*-test.”

- We added all the exact *p*-values for the supplementary Tables 2; 3; 4; 5; 6; 7; 8; 9; 10; 11; 12; 13; 14, and in Supplementary Figure 3c.
- We created a DOI for our code repository, and added the reference to the *Code availability* section (reference 146).
- We added references to the missing R libraries in the main manuscript (dplyr, oro.nifti).
- Typos:
 - *Statistical Analyses* sub-section (*Method* section): “(R version: 4.1.0)”